# NEURAL STRUCTURED PREDICTION FOR INDUCTIVE NODE CLASSIFICATION

**Meng Qu**[*1,2], **Huiyu Cai**[*1,2], **Jian Tang**[1,3,4]
[1]Mila - Québec AI Institute
[2]Université de Montréal
[3]HEC Montréal
[4]Canadian Institute for Advanced Research (CIFAR)

## ABSTRACT

This paper studies node classification in the inductive setting, i.e., aiming to learn a model on labeled training graphs and generalize it to infer node labels on unlabeled test graphs. This problem has been extensively studied with graph neural networks (GNNs) by learning effective node representations, as well as traditional structured prediction methods for modeling the structured output of node labels, e.g., conditional random fields (CRFs). In this paper, we present a new approach called the Structured Proxy Network (SPN), which combines the advantages of both worlds. SPN defines flexible potential functions of CRFs with GNNs. However, learning such a model is nontrivial as it involves optimizing a maximin game with high-cost inference. Inspired by the underlying connection between joint and marginal distributions defined by Markov networks, we propose to solve an approximate version of the optimization problem as a proxy, which yields a near-optimal solution, making learning more efficient. Extensive experiments on two settings show that our approach outperforms many competitive baselines [1].

## 1 INTRODUCTION

Graph-structured data are ubiquitous in the real world, covering a variety of applications. This paper studies node classification, a fundamental problem in the machine learning community. Most existing efforts focus on the transductive setting (Kipf & Welling, 2017; Veličković et al., 2018), i.e., using a small set of labeled nodes in a graph to classify the rest of nodes. In this paper, we study node classification in the inductive setting (Hamilton et al., 2017), which is receiving growing interest. Given some training graphs with all nodes labeled, we aim to classify nodes in unlabeled test graphs.

This problem has been recently studied with graph neural networks (GNNs) (Kipf & Welling, 2017; Hamilton et al., 2017; Gilmer et al., 2017; Veličković et al., 2018). GNNs infer the marginal label distribution of each node by learning useful node representations based on node features and edges. Once a GNN is learned on training graphs, it can be further applied to test graphs to infer node labels. Owing to the high capacity of nonlinear neural architectures, GNNs achieve impressive results on many datasets. However, one limitation of GNNs is that they ignore the joint dependency of node labels, and therefore node labels are predicted separately without modeling structured output.

Indeed, modeling structured output has been widely explored by the literature of structured prediction (BakIr et al., 2007). Structured prediction methods predict node labels collectively, so the label prediction of each node can be improved according to the predicted labels of neighboring nodes. One representative approach is the conditional random field (CRF) (Lafferty et al., 2001). A CRF models the joint distribution of node labels with Markov networks, and thus training CRFs becomes a learning task in graphical models, while predicting node labels corresponds to an inference task. Typically, the potential functions in CRFs are parameterized as log-linear functions, which suffer from low model capacities. One remedy for this is to define potential functions with GNNs (Ma et al., 2018; Qu et al., 2019). However, most of the effective methods for learning CRFs involve a

---

[*]Equal contribution.
[1]Codes are available at `https://github.com/DeepGraphLearning/SPN`.

maximin game (Wainwright & Jordan, 2008; Sutton & McCallum, 2012), making learning often hard to converge, especially when GNNs are used to parameterize potential functions. Besides, as learning CRFs requires doing inference on the graphical models, the combined model requires a long run time.

In this paper, we address these challenges by proposing SPN (Structured Proxy Network), which is high in capacity, efficient in learning, and able to model the joint dependency of node labels. SPN is inspired by theoretical works in graphical models (Wainwright & Jordan, 2008), which reveal close connections between the joint label distribution and the node/edge marginal label distribution in a Markov network. Based on that, we approximate the original optimization problem with a proxy problem, where the potential functions in CRFs are defined by combining a collection of node/edge pseudomarginal distributions, which are parameterized by GNNs that satisfy a few simple constraints. This proxy problem can be easily solved by maximizing the data likelihood on each node and edge, which yields a near-optimal joint label distribution on training graphs. Once the model is learned, we apply it to test graphs and run loopy belief propagation (Murphy et al., 1999) to infer node labels. Experiments on two settings against both GNNs and CRFs prove the effectiveness of our approach.

Note that although SPN is tested on inductive node classification, this method is quite general and can be applied to many other structured prediction tasks as well, such as POS tagging (Church, 1988) and named entity recognition (Sang & De Meulder, 2003). Please refer to Sec. 4.3 for more details.

## 2 RELATED WORK

Graph neural networks (GNNs) perform node classification by learning useful node representations (Kipf & Welling, 2017; Gilmer et al., 2017; Veličković et al., 2018). Most earlier efforts focus on designing GNNs for transductive node classification (Yang et al., 2016; Gao & Ji, 2019; Xhonneux et al., 2020), and many recent works move to the inductive setting (Hamilton et al., 2017; Gao et al., 2018; Chiang et al., 2019; Li et al., 2019; Chen et al., 2020a; Zeng et al., 2020). Because of high capacity and efficient training, GNNs achieve impressive results on inductive node classification. Despite the success, GNNs only try to model the marginal distribution of each node label and predict node labels separately without considering joint dependency. In contrast, SPN models joint distributions of node labels with CRFs, which predicts node labels collectively to improve results.

Another type of approach for inductive node classification is structured prediction, which focuses on modeling the dependency of node labels, so that the predicted node labels are more consistent. One representative approach is structured SVM (Tsochantaridis et al., 2005; Finley & Joachims, 2008; Sarawagi & Gupta, 2008), but it lacks a probabilistic interpretation to handle the uncertainty of the prediction. Another representative probabilistic approach is conditional random field (Lafferty et al., 2001; Sutton & McCallum, 2006), which models the distribution of output spaces by using a Markov network. CRFs have been proven effective in many applications, such as POS tagging (Lafferty et al., 2001), shallow parsing (Sha & Pereira, 2003), image labeling (He et al., 2004), and sequence labeling (Lample et al., 2016; Ma & Hovy, 2016; Liu et al., 2018). Nevertheless, the potential functions in CRFs are typically defined as log-linear functions, suffering from low model capacity.

There are also some recent works trying to combine GNNs and CRFs. Some works use GNNs to solve inference problems in graphical models (Dai et al., 2016; Satorras et al., 2019; Zhang et al., 2020; Chen et al., 2020b; Satorras & Welling, 2020). In contrast, our approach uses GNNs to parameterize the potential functions in CRFs, which is in a similar vein to Ma et al. (2018); Qu et al. (2019); Ma et al. (2019; 2021); Wang et al. (2021). Among them, Ma et al. (2018) and Qu et al. (2019) optimize the pseudolikelihood (Besag, 1975) for model learning, and Wang et al. (2021) optimizes a cross-entropy loss on each single node, which can yield poor approximation of the true joint likelihood (Koller & Friedman, 2009; Sutton & McCallum, 2012). Our approach instead solves a proxy problem, which yields a near-optimal solution to the original problem of maximizing likelihood, and thus gets superior results. For Ma et al. (2019) and Ma et al. (2021), they focus on transductive node classification and continuous labels respectively, which are different from our work.

Lastly, learning CRFs has also been widely studied. Some works solve a maximin game as a surrogate for learning (Sutton & McCallum, 2012) and some others maximize a lower bound of the likelihood function (Sutton & McCallum, 2009). However, these maximin games are often hard to optimize and the lower bounds are often loose. Different from them, we follow Wainwright et al. (2003) and build an approximate optimization problem as a proxy, which is easier to solve and yields better results.

## 3 PRELIMINARY

This paper focuses on inductive node classification (Hamilton et al., 2017), a fundamental problem in both graph machine learning and structured prediction. We employ a probabilistic formalization for the problem with some labeled training graphs and unlabeled test graphs. Each training graph is given as $(\mathbf{y}_V^*, \mathbf{x}_V, E)$, where $\mathbf{x}_V$ and $\mathbf{y}_V^*$ are features and labels of a set of nodes $V$, and $E$ is a set of edges. For each test graph $(\mathbf{x}_{\tilde{V}}, \tilde{E})$, only features $\mathbf{x}_{\tilde{V}}$ and edges $\tilde{E}$ are given. Then we aim to solve:

- **Learning.** On training graphs, learn a probabilistic model to approximate $p(\mathbf{y}_V|\mathbf{x}_V, E)$.
- **Inference.** For each test graph, infer node labels $\mathbf{y}_{\tilde{V}}^*$ according to the distribution $p(\mathbf{y}_{\tilde{V}}|\mathbf{x}_{\tilde{V}}, \tilde{E})$.

The problem has been extensively studied in both graph machine learning and structured prediction fields, and representative methods are GNNs and CRFs respectively. Next, we introduce the details.

### 3.1 GRAPH NEURAL NETWORKS

For inductive node classification, graph neural networks (GNNs) learn node representations to predict marginal label distributions of nodes. GNNs assume all node labels are independent conditioned on node features and edges, so the joint label distribution is factorized into a set of marginals as below:

$$p_\theta(\mathbf{y}_V|\mathbf{x}_V, E) = \prod_{s \in V} p_\theta(y_s|\mathbf{x}_V, E). \tag{1}$$

Each marginal distribution $p_\theta(y_s|\mathbf{x}_V, E)$ is modeled as a categorical distribution over label candidates, and the label probabilities are computed by applying a linear softmax classifier to the representation of node $s$. In general, node representations are learned via the message passing mechanism (Gilmer et al., 2017), which brings high capacity to GNNs. Also, owing to the factorization in Eq. (1), learning and inference can be easily solved in GNNs, where we simply need to compute loss and make prediction on each node separately. However, GNNs approximate only the marginal label distributions of nodes on training graphs, which may generalize badly and result in poor approximation of node marginal label distributions on test graphs. Also, the labels of different nodes are separately predicted according to their own marginal label distributions, yet the joint dependency of node labels is ignored.

### 3.2 CONDITIONAL RANDOM FIELDS

For inductive node classification, conditional random fields (CRFs) build graphical models for node classification. A popular model is the pair-wise CRF, which formalizes the joint label distribution as:

$$p_\theta(\mathbf{y}_V|\mathbf{x}_V, E) = \frac{1}{Z_\theta(\mathbf{x}_V, E)} \exp\{\sum_{s \in V} \theta_s(y_s, \mathbf{x}_V, E) + \sum_{(s,t) \in E} \theta_{st}(y_s, y_t, \mathbf{x}_V, E)\} \tag{2}$$

where $Z_\theta(\mathbf{x}_V, E)$ is the partition function. $\theta_s(y_s, \mathbf{x}_V, E)$ and $\theta_{st}(y_s, y_t, \mathbf{x}_V, E)$ are scalar scores contributed by each node $s$ and each edge $(s, t)$. In practice, these $\theta$-functions can be either defined as simple linear functions or complicated GNNs. To make the notation concise, we will omit $\mathbf{x}_V$ and $E$ in the $\theta$-functions, e.g., simplifying $\theta_s(y_s, \mathbf{x}_V, E)$ as $\theta_s(y_s)$. With these $\theta$-functions, CRFs are able to model the joint dependency of node labels and therefore achieve structured prediction.

However, learning CRFs to maximize likelihood $p_\theta(\mathbf{y}_V^*|\mathbf{x}_V, E)$ on training graphs is nontrivial in general, as the partition function $Z_\theta(\mathbf{x}_V, E)$ is typically intractable in graphs with loops. Thus, a major line of research instead optimizes a maximin game equivalent to likelihood maximization (Wainwright & Jordan, 2008). The maximin game for each training graph $(\mathbf{y}_V^*, \mathbf{x}_V, E)$ is formalized as follows:

$$\max_\theta \log p_\theta(\mathbf{y}_V^*|\mathbf{x}_V, E) = \max_\theta \min_q \mathcal{L}(\theta, q), \quad \text{with} \quad \mathcal{L}(\theta, q) =$$

$$\sum_{s \in V} \{\theta_s(y_s^*) - \mathbb{E}_{q_s(y_s)}[\theta_s(y_s)]\} + \sum_{(s,t) \in E} \{\theta_{st}(y_s^*, y_t^*) - \mathbb{E}_{q_{st}(y_s, y_t)}[\theta_{st}(y_s, y_t)]\} - H[q(\mathbf{y}_V)]. \tag{3}$$

Here, $q(\mathbf{y}_V)$ is a variational distribution on node labels, $q_s(y_s)$ and $q_{st}(y_s, y_t)$ are its marginal distributions on nodes and edges. $H[q(\mathbf{y}_V)] := -\mathbb{E}_{q(\mathbf{y}_V)}[\log q(\mathbf{y}_V)]$ is the entropy of $q(\mathbf{y}_V)$. Given the maximin game, $q$ and $\theta$ can be alternatively optimized via coordinate descent (Sutton & McCallum, 2012). In each iteration, we first update the node and edge marginals $\{q_s(y_s)\}_{s \in V}$, $\{q_{st}(y_s, y_t)\}_{(s,t) \in E}$

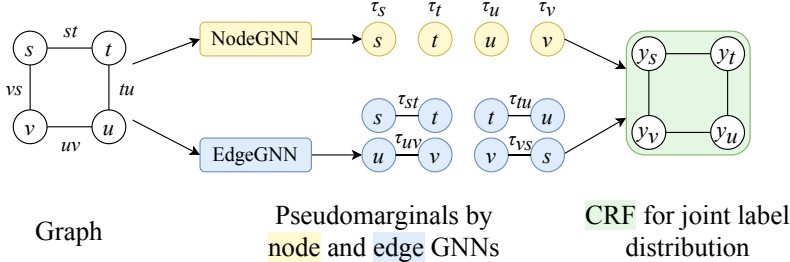

Figure 1: Framework overview of the SPN. Our approach formulates a proxy optimization problem for learning, which is much easier to solve. Given a graph, a node GNN and an edge GNN are used to predict the pseudomarginal label distributions on each node and each edge respectively. Then these pseudomarginals serve as building blocks to construct a near-optimal joint label distribution.

towards those defined by $p_\theta$. This can be done by MCMC, but the time cost is high, so approximate inference is often used, such as loopy belief propagation (Murphy et al., 1999). After $q$ is optimized, we further update $\theta$-functions with the node and edge marginals defined by $q$ via gradient descent.

The optimal $\theta$-functions are characterized by the following moment-matching conditions:

$$p_\theta(y_s|\mathbf{x}_V, E) = \mathbb{I}_{y_s^*}\{y_s\} \quad \forall s \in V, \qquad p_\theta(y_s, y_t|\mathbf{x}_V, E) = \mathbb{I}_{(y_s^*, y_t^*)}\{(y_s, y_t)\} \quad \forall (s, t) \in E, \quad (4)$$

where $\mathbb{I}_a\{b\}$ is an indicator function whose value is 1 if $a = b$ and 0 otherwise. See Sec. A and Sec. B in appendix for detailed derivation of the maximin game as well as the moment-matching conditions.

Once the $\theta$-functions are learned, they can be further applied to each test graph $(\mathbf{x}_{\tilde{V}}, \tilde{E})$ to predict the joint label distribution as $p_\theta(\mathbf{y}_{\tilde{V}}|\mathbf{x}_{\tilde{V}}, \tilde{E})$. Then the best label assignment $\mathbf{y}_{\tilde{V}}^*$ can be inferred by using approximate inference algorithms, such as loopy belief propagation (Murphy et al., 1999).

The major challenge of CRFs lies in learning. On the one hand, learning relies on inference, meaning that we have to update $\{q_s(y_s)\}_{s \in V}$, $\{q_{st}(y_s, y_t)\}_{(s,t) \in E}$ to approximate the node and edge marginals of $p_\theta$ at each step, which can be expensive. On the other hand, as learning involves a maximin game and the optimal $q$ of the inner minimization problem in Eq. (3) is intractable, we can only maximize an upper bound of the likelihood function for $\theta$, making learning unstable. The problem becomes even more severe when $\theta$ is parameterized by highly nonlinear neural models, e.g. GNNs.

## 4 MODEL

In this section, we introduce our proposed approach Structured Proxy Network (SPN). The general idea of SPN is to combine GNNs and CRFs by parameterizing potential functions in CRFs with GNNs, and therefore SPN enjoys high capacity and can model the joint dependency of node labels.

However, as elaborated in Sec. 3.2, learning such a model on training graphs is challenging due to the maximin game in optimization. Inspired by the connection between the joint and marginal distributions of CRFs, we instead construct a new optimization problem, which serves as a proxy for model learning. Compared with the original maximin game, the proxy problem is much easier to solve, where we can simply train two GNNs to approximate the marginal label distributions on nodes and edges, and further combine these pseudomarginals (defined in Prop. 1) into a near-optimal joint label distribution. This joint label distribution can be further refined by optimizing the maximin game, although it is optional and often unnecessary, as this distribution is often close enough to the optimal one. With this proxy problem for model learning, learning becomes more stable and efficient.

Afterwards, the learned model is used to predict the joint label distribution on test graphs. Then we run loopy belief propagation to infer node labels. Now, we introduce the details of our approach.

### 4.1 LEARNING

The learning task aims at training $\theta$ to maximize the log-likelihood function $\log p_\theta(\mathbf{y}_V^*|\mathbf{x}_V, E)$ for each training graph $(\mathbf{y}_V^*, \mathbf{x}_V, E)$, which is highly challenging. Therefore, instead of directly optimizing this goal, we solve an approximate version of the problem as a proxy, which is training a node GNN and an edge GNN to maximize the log-likelihood of observed labels on nodes and edges.

**The Proxy Problem.** The proxy problem is inspired by Wainwright & Jordan (2008), which points out that the marginal label distributions on nodes and edges defined by a Markov network have inherent connections with the joint distribution. This connection is stated in the proposition below.

**Proposition 1** *Consider a set of nonzero pseudomarginals $\{\tau_s(y_s)\}_{s \in V}$ and $\{\tau_{st}(y_s, y_t)\}_{(st) \in E}$ which satisfy $\sum_{y_s} \tau_{st}(y_s, y_t) = \tau_t(y_t)$ and $\sum_{y_t} \tau_{st}(y_s, y_t) = \tau_s(y_s)$ for all $(s, t) \in E$.*

*If we parameterize the $\theta$-functions of $p_\theta$ in Eq. (2) in the following way:*

$$\theta_s(y_s) = \log \tau_s(y_s) \quad \forall s \in V, \qquad \theta_{st}(y_s, y_t) = \log \frac{\tau_{st}(y_s, y_t)}{\tau_s(y_s)\tau_t(y_t)} \quad \forall (s, t) \in E, \tag{5}$$

*then $\{\tau_s(y_s)\}_{s \in V}$ and $\{\tau_{st}(y_s, y_t)\}_{(s,t) \in E}$ are specified by a fixed point of the sum-product loopy belief propagation algorithm when applied to the joint distribution $p_\theta$, which implies that:*

$$\tau_s(y_s) \approx p_\theta(y_s) \quad \forall s \in V, \qquad \tau_{st}(y_s, y_t) \approx p_\theta(y_s, y_t) \quad \forall (s, t) \in E. \tag{6}$$

The proof is provided in Sec. C. With the proposition, we observe that if we parameterize the $\theta$-functions by combining a set of pseudomarginals $\{\tau_s(y_s)\}_{s \in V}$ and $\{\tau_{st}(y_s, y_t)\}_{(s,t) \in E}$ in the way defined by Eq. (5), then those pseudomarginals can well approximate the true marginals of the joint distribution $p_\theta$, i.e., $\tau_s(y_s) \approx p_\theta(y_s)$ and $\tau_{st}(y_s, y_t) \approx p_\theta(y_s, y_t)$ for all nodes $s$ and edges $(s, t)$. Given this precondition, if we further have $\tau_s(y_s) \approx \mathbb{I}_{y_s^*}\{y_s\}$ and $\tau_{st}(y_s, y_t) \approx \mathbb{I}_{(y_s^*, y_t^*)}\{(y_s, y_t)\}$, then the moment-matching conditions in Eq. (4) for the optimal $\theta$-functions are roughly satisfied. This implies the joint distribution $p_\theta(\mathbf{y}_V | \mathbf{x}_V, E)$ derived in this way is a near-optimal one.

With the observation, rather than directly using GNNs to parameterize the $\theta$-functions, we use a node GNN and an edge GNN to parameterize the pseudomarginals $\{\tau_s(y_s)\}_{s \in V}$ and $\{\tau_{st}(y_s, y_t)\}_{(s,t) \in E}$. For the pseudomarginal $\tau_s(y_s)$ on node $s$, we apply the node GNN to node features $\mathbf{x}_V$ and edges $E$, yielding a representation $\mathbf{u}_s$ for node $s$. Then we apply a softmax classifier to $\mathbf{u}_s$ to compute $\tau_s(y_s)$:

$$\{\mathbf{u}_s\}_{u \in V} = \text{GNN}_{\text{node}}(\mathbf{x}_V, E), \qquad \tau_s(y_s) = \text{softmax}(f(\mathbf{u}_s))[y_s], \tag{7}$$

where $f$ maps a node representation to a $|\mathcal{Y}|$-dimensional logit and $\mathcal{Y}$ is the node label set. Similarly, we apply the edge GNN to compute a representation $\mathbf{v}_s$ for each node $s$, and model $\tau_{st}(y_s, y_t)$ as:

$$\{\mathbf{v}_s\}_{s \in V} = \text{GNN}_{\text{edge}}(\mathbf{x}_V, E) \qquad \tau_{st}(y_s, y_t) = \text{softmax}(g(\mathbf{v}_s, \mathbf{v}_t))[y_s, y_t], \tag{8}$$

where $g$ is a function mapping a pair of representations to a $(|\mathcal{Y}| \times |\mathcal{Y}|)$-dimensional logit.

Given the parameterization, we construct the following problem as a proxy for learning $\theta$-functions:

$$\min_{\tau, \theta} \sum_{s \in V} d\left(\mathbb{I}_{y_s^*}\{y_s\}, \tau_s(y_s)\right) + \sum_{(s,t) \in E} d\left(\mathbb{I}_{(y_s^*, y_t^*)}\{(y_s, y_t)\}, \tau_{st}(y_s, y_t)\right),$$

$$\text{subject to} \quad \theta_s = \log \tau_s(y_s), \quad \theta_{st}(y_s, y_t) = \log \frac{\tau_{st}(y_s, y_t)}{\tau_s(y_s)\tau_t(y_t)}, \tag{9}$$

$$\text{and} \quad \sum_{y_s} \tau_{st}(y_s, y_t) = \tau_t(y_t), \quad \sum_{y_t} \tau_{st}(y_s, y_t) = \tau_s(y_s),$$

for all nodes and edges, where $d$ can be any divergence measure between two distributions. By solving the above problem, $\{\tau_s(y_s)\}_{s \in V}$ and $\{\tau_{st}(y_s, y_t)\}_{(s,t) \in E}$ will be valid pseudomarginals which can well approximate the true labels, i.e., $\tau_s(y_s) \approx \mathbb{I}_{y_s^*}\{y_s\}$ and $\tau_{st}(y_s, y_t) \approx \mathbb{I}_{(y_s^*, y_t^*)}\{(y_s, y_t)\}$. Then according to the constraint in the second line of Eq. (9), $\theta$-functions are formed in a way to enable $\tau_s(y_s) \approx p_\theta(y_s)$ and $\tau_{st}(y_s, y_t) \approx p_\theta(y_s, y_t)$ as stated in the Prop. 1. Combining these two sets of formula results in $p_\theta(y_s) \approx \mathbb{I}_{y_s^*}\{y_s\}$ and $p_\theta(y_s, y_t) \approx \mathbb{I}_{y_s^*}\{y_s\}$. We see that the moment-matching conditions in Eq. (4) for the optimal joint label distribution are roughly achieved, implying that the derived joint distribution $p_\theta(\mathbf{y}_V | \mathbf{x}_V, E)$ is a near-optimal solution to the original learning problem.

One good property of the proxy problem is that it can be solved easily. The last consistency constraint (i.e. $\sum_{y_s} \tau_{st}(y_s, y_t) = \tau_t(y_t)$ and $\sum_{y_t} \tau_{st}(y_s, y_t) = \tau_s(y_s)$) can be ignored during optimization, since by optimizing the objective function, the optimal pseudomarginals $\tau$ should well approximate the observed node and edge marginals, i.e., $\tau_s(y_s) \approx \mathbb{I}_{y_s^*}\{y_s\}$ and $\tau_{st}(y_s, y_t) \approx \mathbb{I}_{(y_s^*, y_t^*)}\{(y_s, y_t)\}$, and hence $\tau$ will almost naturally satisfy the consistency constraint. We also tried some constrained

optimization methods to handle the consistency constraint, but they yield no improvement. See Sec. D of appendix for more details. Thus, we can simply train the pseudomarginals parameterized by GNNs to approximate the true node and edge labels on training graphs, i.e., minimizing $d(\mathbb{I}_{y_s^*}\{y_s\}, \tau_s(y_s))$ and $d(\mathbb{I}_{(y_s^*, y_t^*)}\{(y_s, y_t)\}, \tau_{st}(y_s, y_t))$. Then we build $\theta$-functions as in Eq. (5) to obtain a near-optimal joint distribution. In practice, we choose $d$ to be the KL divergence, yielding an objective for $\tau$ as:

$$\max_\tau \sum_{s \in V} \log \tau_s(y_s^*) + \sum_{(s,t) \in E} \log \tau_{st}(y_s^*, y_t^*). \tag{10}$$

This objective function is very intuitive, where we simply try to optimize the node GNN and edge GNN to maximize the log-likelihood function of the observed labels on nodes and edges.

**Refinement.** By solving the proxy problem, we can obtain a near-optimal joint distribution. In practice, we observe that when we have a large amount of training data, further refining this joint distribution by solving the maximin game in Eq. (3) for a few iterations can lead to further improvement. Formally, each iteration of refinement has two steps. In the first step, we run sum-product loopy belief propagation (Murphy et al., 1999), which yields a collection of node and edge marginals (i.e., $\{q_s(y_s)\}_{s \in V}$ and $\{q_{st}(y_s, y_t)\}_{(s,t) \in E}$) as approximation to the marginals defined by $p_\theta$. In the second step, we update the $\theta$-functions parameterized by the node and edge GNNs to maximize:

$$\sum_{s \in V} \left\{ \theta_s(y_s^*) - \mathbb{E}_{q_s(y_s)}[\theta_s(y_s)] \right\} + \sum_{(s,t) \in E} \left\{ \theta_{st}(y_s^*, y_t^*) - \mathbb{E}_{q_{st}(y_s, y_t)}[\theta_{st}(y_s, y_t)] \right\}. \tag{11}$$

Intuitively, we treat the true label $y_s^*$ and $(y_s^*, y_t^*)$ of each node and edge as positive examples, and encourage the $\theta$-functions to raise up their scores. Meanwhile, those labels sampled from $q_s(y_s)$ and $q_{st}(y_s, y_t)$ act as negative examples, and the $\theta$-functions are updated to decrease their scores.

## 4.2 INFERENCE

After learning, we apply the node and edge GNNs to each test graph $(\mathbf{x}_{\tilde{V}}, \tilde{E})$ to compute the $\theta$-functions, which are integrated into an approximate joint label distribution $p_\theta(\mathbf{y}_{\tilde{V}}|\mathbf{x}_{\tilde{V}}, \tilde{E})$. Then we use this distribution to infer the best label $\mathbf{y}_{\tilde{s}}^*$ for each node $\tilde{s} \in \tilde{V}$, where two settings are considered.

**Node-level Accuracy.** Typically, we care about the node-level accuracy, i.e., how likely we can correctly classify a node in test graphs. Intuitively, the best label $y_{\tilde{s}}^*$ for each test node $\tilde{s} \in \tilde{V}$ should be predicted as $y_{\tilde{s}}^* = \arg\max_{y_{\tilde{s}}} p_\theta(y_{\tilde{s}}|\mathbf{x}_{\tilde{V}}, \tilde{E})$, where $p_\theta(y_{\tilde{s}}|\mathbf{x}_{\tilde{V}}, \tilde{E})$ is the marginal label distribution of node $\tilde{s}$ induced by the joint $p_\theta(\mathbf{y}_{\tilde{V}}|\mathbf{x}_{\tilde{V}}, \tilde{E})$. In practice, the exact marginal is intractable, so we apply loopy belief propagation (Murphy et al., 1999) for approximate inference. For each edge $(\tilde{s}, \tilde{t})$ in test graphs, we introduce a message function $m_{\tilde{t} \to \tilde{s}}(y_{\tilde{s}})$ and iteratively update all messages as:

$$m_{\tilde{t} \to \tilde{s}}(y_{\tilde{s}}) \propto \sum_{y_{\tilde{t}}} \{ \exp(\theta_{\tilde{t}}(y_{\tilde{t}}) + \theta_{\tilde{s}\tilde{t}}(y_{\tilde{s}}, y_{\tilde{t}})) \prod_{\tilde{s}' \in N(\tilde{t}) \setminus \tilde{s}} m_{\tilde{s}' \to \tilde{t}}(y_{\tilde{t}}) \}, \tag{12}$$

where $N(\tilde{s})$ denotes the set of neighboring nodes for node $\tilde{s}$. Once the above process converges or after sufficient iterations, the label of each node $\tilde{s}$ can be inferred in the following way:

$$y_{\tilde{s}}^* = \arg\max_{y_{\tilde{s}}} [\exp(\theta_{\tilde{s}}(y_{\tilde{s}})) \prod_{\tilde{t} \in N(\tilde{s})} m_{\tilde{t} \to \tilde{s}}(y_{\tilde{s}})]. \tag{13}$$

**Graph-level Accuracy.** In some other cases, we might care about the graph-level accuracy, i.e., how likely we can correctly classify all nodes in a given test graph. In this case, the best prediction of node labels is given by $\mathbf{y}_{\tilde{V}}^* = \arg\max_{\mathbf{y}_{\tilde{V}}} p(\mathbf{y}_{\tilde{V}}|\mathbf{x}_{\tilde{V}}, \tilde{E})$. This problem can be approximately solved by the max-product variant of loopy belief propagation, which simply replaces the sum over $y_{\tilde{t}}$ in Eq. (12) with max (Weiss & Freeman, 2001). Afterwards, the best node label can be still decoded via Eq. (13).

## 4.3 DISCUSSION

In practice, many structured prediction problems can be viewed as special cases of inductive node classification, where the graphs between nodes have some special structures. For example in sequence labeling tasks (e.g., named entity recognition), the graphs between nodes have sequential structures. Thus, SPN can be applied to these tasks as well. In order for better results, one might replace GNNs with other neural models which are specifically designed for the studied task to better estimate the pseudomarginals. For example in sequence labeling tasks, recurrent neural networks can be used.

## 5 Experiment

### 5.1 Datasets

We consider datasets in two settings, which focus on node-level and graph-level accuracy respectively.

**Node-level Accuracy.** The node-level accuracy measures *how likely a model can predict the correct label of a node in test graphs*. We use the **PPI** dataset (Zitnik & Leskovec, 2017; Hamilton et al., 2017), which has 20 training graphs. To make the dataset more challenging, we also try using only the first 1/2/10 training graphs, yielding another three datasets **PPI-1**, **PPI-2**, and **PPI-10**. Besides, we also build a **DBLP** dataset from the citation network in Tang et al. (2008). Papers from eight conferences are treated as nodes, and we split them into three categories for classification according to conference domains [2]. For each paper, we compute the mean GloVe embedding (Pennington et al., 2014) of words in the title and abstract as node features. The training/validation/test graph is formed as the citation graph of papers published before 1999, from 2000 to 2009, after 2010 respectively.

**Graph-level Accuracy.** The graph-level accuracy measures *how likely a model can correctly classify all the nodes for a given test graph*. We construct three datasets from the Cora, Citeseer, and Pubmed datasets used for transductive node classification (Yang et al., 2016). Each raw dataset has a single graph. For each training/validation/test node of the raw dataset, we treat its ego network [3] as a training/validation/test graph. We denote the datasets as **Cora\***, **Citeseer\***, **Pubmed\***.

### 5.2 Compared Algorithms

**Graph Neural Networks.** For GNNs, we choose a few well-known model architectures for comparison, including GCN (Kipf & Welling, 2017), GraphSage (Hamilton et al., 2017), GAT (Veličković et al., 2018), Graph U-Net (Gao & Ji, 2019) and GCNII (Chen et al., 2020a).

**Conditional Random Fields.** For CRFs, we consider three variants. (1) *CRF-linear*. This variant uses linear $\theta$-functions in Eq. (2), which takes the features on nodes and edges for computation. (2) *CRF-GNN*. This variant parameterizes the $\theta$-functions as $\theta_s(y_s) = f(\mathbf{u}_s)$ and $\theta_{st}(y_s, y_t) = g(\mathbf{v}_s, \mathbf{v}_t)$, with $f$ and $g$ defined in Eq. (7) and Eq. (8), where the node representations are generated by different GNN architectures (e.g., CRF-GAT). We train these models via the maximin game as in Eq. (3) with sum-product loopy belief propagation. (3) *GMNN*. We also consider GMNN (Qu et al., 2019), an approach combining GNNs and CRFs, which optimizes the pseudolikelihood function for learning.

**Our Approach.** For SPNs, we try different GNN architectures for defining the node and edge GNNs (e.g., SPN-GAT). By default, we only solve the proxy problem without performing refinement. We systematically compare the results with and without refinement in part 2 of Sec. 5.5.

### 5.3 Evaluation Metrics

On Cora\*, Citeseer\*, and Pubmed\*, we report the percentage of test graphs where all the nodes are correctly classified (i.e., graph-level accuracy). On DBLP and PPI, we report accuracy and micro-F1 based on the percentage of test nodes which are correctly classified (i.e., node-level accuracy). For Cora\*, Citeseer\*, and Pubmed\*, we run each compared method with 10 different seeds to report the mean accuracy and the standard deviation. For DBLP and PPI, we run each method with 5 seeds.

### 5.4 Experimental Setup

For GNNs, by default we use the same architectures (e.g., number of neurons, number of layers) as used in the original papers. Adam (Kingma & Ba, 2015) is used for training. For the edge GNN in Eq. (8), we add a hyperparameter $\gamma$ to control the annealing temperature of the logit $g(\mathbf{v}_s, \mathbf{v}_t)$ before the softmax function during belief propagation. Empirically, we find that max-product belief propagation works better than the sum-product variant in most cases, so we use the max-product version by default. By default, we do not run refinement when training SPNs. See Sec. F for details.

---

[2]ML: ICML/NeurIPS. CV: ICCV/CVPR/ECCV. NLP: ACL/EMNLP/NAACL.

[3]The local subgraph formed by a node and its direct neighbors.

Table 1: Result on PPI datasets (in %). SPNs get consistent improvement on GNNs and CRFs.

| Algorithm | PPI-1 | | PPI-2 | | PPI-10 | | PPI | |
|---|---|---|---|---|---|---|---|---|
| | Accuracy | Micro-F1 | Accuracy | Micro-F1 | Accuracy | Micro-F1 | Accuracy | Micro-F1 |
| GCN | $76.62 \pm 0.10$ | $54.55 \pm 0.29$ | $77.48 \pm 0.12$ | $56.10 \pm 0.36$ | $80.43 \pm 0.10$ | $62.48 \pm 0.27$ | $82.28 \pm 0.24$ | $66.52 \pm 0.89$ |
| GraphSAGE | $81.02 \pm 0.07$ | $67.30 \pm 0.11$ | $84.13 \pm 0.04$ | $72.93 \pm 0.04$ | $95.34 \pm 0.03$ | $92.18 \pm 0.05$ | $98.51 \pm 0.02$ | $97.51 \pm 0.03$ |
| GAT | $77.49 \pm 0.20$ | $60.72 \pm 0.25$ | $81.35 \pm 0.19$ | $68.55 \pm 0.30$ | $96.14 \pm 0.15$ | $93.53 \pm 0.24$ | $98.85 \pm 0.05$ | $98.06 \pm 0.08$ |
| Graph U-Net | $77.17 \pm 0.07$ | $55.54 \pm 0.33$ | $78.22 \pm 0.04$ | $59.12 \pm 0.30$ | $83.15 \pm 0.04$ | $68.70 \pm 0.08$ | $86.29 \pm 0.04$ | $75.57 \pm 0.18$ |
| GCNII | $80.99 \pm 0.07$ | $65.79 \pm 0.25$ | $84.81 \pm 0.06$ | $74.54 \pm 0.14$ | $97.53 \pm 0.01$ | $95.86 \pm 0.01$ | $99.39 \pm 0.00$ | $98.97 \pm 0.00$ |
| CRF-linear | $65.33 \pm 2.77$ | $48.30 \pm 0.35$ | $67.20 \pm 2.24$ | $49.45 \pm 0.97$ | $69.72 \pm 0.65$ | $50.17 \pm 0.39$ | $69.98 \pm 0.30$ | $50.61 \pm 0.35$ |
| CRF-GCN | $76.33 \pm 0.21$ | $50.79 \pm 0.74$ | $76.27 \pm 0.10$ | $49.47 \pm 0.63$ | $77.08 \pm 0.07$ | $52.36 \pm 0.72$ | $77.34 \pm 0.07$ | $53.60 \pm 0.36$ |
| CRF-GraphSAGE | $77.43 \pm 0.28$ | $54.57 \pm 1.07$ | $77.25 \pm 0.36$ | $53.48 \pm 1.00$ | $77.65 \pm 0.38$ | $54.44 \pm 1.34$ | $77.21 \pm 0.19$ | $54.50 \pm 3.09$ |
| CRF-GAT | $76.50 \pm 0.49$ | $52.95 \pm 0.40$ | $76.76 \pm 0.61$ | $55.01 \pm 0.93$ | $74.58 \pm 0.92$ | $54.98 \pm 1.13$ | $70.42 \pm 0.72$ | $53.27 \pm 0.42$ |
| CRF-GCNII | $79.98 \pm 0.32$ | $61.22 \pm 1.10$ | $81.73 \pm 0.33$ | $66.37 \pm 0.56$ | $92.11 \pm 0.28$ | $87.10 \pm 0.40$ | $96.94 \pm 0.12$ | $94.95 \pm 0.19$ |
| GMNN | $77.55 \pm 0.53$ | $57.20 \pm 2.63$ | $81.21 \pm 0.87$ | $67.46 \pm 2.92$ | $94.67 \pm 2.77$ | $90.72 \pm 5.28$ | $97.00 \pm 2.98$ | $94.69 \pm 5.60$ |
| SPN-GCN | $77.07 \pm 0.05$ | $54.15 \pm 0.17$ | $78.02 \pm 0.05$ | $55.73 \pm 0.15$ | $80.59 \pm 0.04$ | $61.36 \pm 0.11$ | $82.56 \pm 0.20$ | $66.70 \pm 0.77$ |
| SPN-GraphSAGE | $\mathbf{82.11} \pm 0.03$ | $\mathbf{68.56} \pm 0.07$ | $85.40 \pm 0.05$ | $74.45 \pm 0.07$ | $95.28 \pm 0.02$ | $91.99 \pm 0.04$ | $98.55 \pm 0.02$ | $97.56 \pm 0.03$ |
| SPN-GAT | $79.01 \pm 0.17$ | $64.02 \pm 0.40$ | $83.55 \pm 0.12$ | $72.37 \pm 0.18$ | $96.68 \pm 0.13$ | $94.41 \pm 0.21$ | $99.04 \pm 0.06$ | $98.38 \pm 0.10$ |
| SPN-GCNII | $82.01 \pm 0.03$ | $67.80 \pm 0.11$ | $\mathbf{85.83} \pm 0.04$ | $\mathbf{75.96} \pm 0.05$ | $\mathbf{97.55} \pm 0.01$ | $\mathbf{95.87} \pm 0.02$ | $\mathbf{99.41} \pm 0.00$ | $\mathbf{99.02} \pm 0.00$ |

Table 2: Accuracy on Cora*, Citeseer*, Pubmed*, DBLP (in %). SPNs achieve the best result.

| Algorithm | Cora* | Citeseer* | Pubmed* | DBLP |
|---|---|---|---|---|
| GCN | $57.26 \pm 0.66$ | $46.24 \pm 0.61$ | $51.84 \pm 0.45$ | $76.60 \pm 2.32$ |
| GraphSAGE | $49.02 \pm 2.37$ | $41.32 \pm 2.41$ | $48.61 \pm 1.28$ | $73.81 \pm 0.90$ |
| GAT | $51.99 \pm 3.51$ | $47.94 \pm 0.46$ | $50.89 \pm 0.52$ | $79.16 \pm 1.44$ |
| Graph U-Net | $56.07 \pm 0.57$ | $45.91 \pm 1.65$ | $51.77 \pm 0.97$ | $75.21 \pm 2.68$ |
| GCNII | $59.15 \pm 0.67$ | $46.39 \pm 0.92$ | $53.54 \pm 0.98$ | $81.79 \pm 0.88$ |
| CRF-linear | $42.78 \pm 3.94$ | $40.60 \pm 0.81$ | $43.90 \pm 2.91$ | $54.26 \pm 1.27$ |
| CRF-GAT | $49.10 \pm 3.80$ | $42.89 \pm 1.30$ | $47.79 \pm 1.33$ | $59.14 \pm 4.15$ |
| CRF-UNet | $53.49 \pm 2.47$ | $43.66 \pm 2.12$ | $50.02 \pm 0.88$ | $57.46 \pm 3.07$ |
| CRF-GCNII | $36.18 \pm 5.75$ | $38.27 \pm 4.82$ | $41.71 \pm 4.79$ | $60.55 \pm 2.23$ |
| GMNN | $54.30 \pm 1.15$ | $48.46 \pm 1.06$ | $51.70 \pm 1.23$ | $76.54 \pm 2.93$ |
| SPN-GAT | $58.78 \pm 1.21$ | $\mathbf{49.02} \pm 0.78$ | $52.91 \pm 0.54$ | $\mathbf{84.84} \pm 0.73$ |
| SPN-UNet | $58.03 \pm 0.54$ | $46.97 \pm 1.06$ | $53.36 \pm 0.67$ | $80.11 \pm 1.59$ |
| SPN-GCNII | $\mathbf{60.47} \pm 0.49$ | $48.34 \pm 0.50$ | $\mathbf{54.35} \pm 0.64$ | $83.57 \pm 1.33$ |

## 5.5 RESULTS

**1. Comparison with other methods.** The main results in the two settings are presented in Tab. 1 and Tab. 2. Compared against different GNN models, our approach achieves consistent improvement (the relative underperformance of SPN-GCN and SPN-SAGE is related to the capacity of the backbone GNNs and is explained in Sec. G.1) by using these GNNs as backbone networks for approximating marginal label distributions on nodes and edges, which demonstrates SPNs are able to model the structured output of node labels by combining with CRFs, and thus achieve better results.

Besides, SPNs also achieve superior results to CRF-GNNs which are trained by directly solving the maximin game in Eq. (3), as well as GMNN which optimizes the pseudolikelihood function. This observation proves the advantage of our proposed proxy optimization problem for learning CRFs.

Table 3: Run time comparison (in sec).

| Algorithm | DBLP | PPI |
|---|---|---|
| GAT | 23.15 | 460.81 |
| CRF (GAT) | 500.43 | 27136.90 |
| SPN(GAT) | 46.86 | 962.92 |

Table 4: Micro-F1 with and w/o refinement (in %).

| Algorithm | Refine | PPI-2 | PPI-10 | PPI |
|---|---|---|---|---|
| SPN- | w/o | $71.52 \pm 0.21$ | $94.41 \pm 0.21$ | $98.38 \pm 0.10$ |
| GAT | with | $\mathbf{71.58} \pm 0.20$ | $\mathbf{94.63} \pm 0.20$ | $\mathbf{98.68} \pm 0.09$ |
| SPN- | w/o | $\mathbf{73.93} \pm 0.08$ | $91.99 \pm 0.04$ | $97.56 \pm 0.03$ |
| GraphSAGE | with | $73.68 \pm 0.10$ | $\mathbf{92.49} \pm 0.02$ | $\mathbf{97.77} \pm 0.02$ |

**2. Effect of refinement.** By solving the proxy optimization problem in Eq. (9), we can obtain a near-optimal joint label distribution on training graphs, based on which we may optionally refine the distribution with the maximin game in Eq. (3). Next, we study the effect of refinement, and we present the results in Tab. 4. By only solving the proxy problem, our approach already achieves impressive results, showing that the proxy problem can well approximate the original learning problem. Only on datasets with sufficient labeled data (e.g., PPI-10, PPI), refinement leads to some improvement.

**3. Model architecture.** SPN uses a node GNN and an edge GNN for computing node and edge marginals independently. In practice, we can also use a shared GNN for both node and edge marginals. We show results of this variant in Tab. 6, where it also achieves significant improvement over GNNs.

**4. Efficiency comparison.** We have seen SPNs achieve better classification results than GNNs and CRFs. Next, we further compare their efficiency by showing the run time on DBLP and PPI. For PPI,

Table 5: Comparison of learning methods (in %).

| Algorithm | Cora* | Citeseer* | PPI-10 |
|---|---|---|---|
| Maximin Game | $49.10 \pm 3.80$ | $42.89 \pm 1.30$ | $54.98 \pm 1.13$ |
| Pseudolikelihood | $54.30 \pm 1.15$ | $48.46 \pm 1.06$ | $90.72 \pm 5.28$ |
| Proxy Problem | $\mathbf{58.78} \pm 1.21$ | $\mathbf{49.02} \pm 0.78$ | $\mathbf{95.87} \pm 0.02$ |

Table 6: Micro-F1 of model variants (in %).

| Algorithm | PPI-1 | PPI-2 | PPI-10 |
|---|---|---|---|
| GAT | $60.72 \pm 0.25$ | $68.55 \pm 0.30$ | $93.53 \pm 0.24$ |
| SPN-GAT node and edge GNNs | $\mathbf{64.02} \pm 0.40$ | $\mathbf{72.37} \pm 0.18$ | $94.41 \pm 0.21$ |
| SPN-GAT a shared GNN | $63.72 \pm 0.38$ | $70.99 \pm 0.25$ | $\mathbf{95.19} \pm 0.15$ |

which has 121 labels, we only report the training times on a single label. We use GAT as the backbone network for CRFs and SPNs. GAT and CRF are trained for 1000 epochs to ensure convergence. For the SPN, we train the node GNN and edge GNN for node/edge classification as in Eq. (10) with 1000 epochs. The run times are presented in Tab. 3. SPNs take twice as long for training than GAT, as a SPN needs to train a node GNN and an edge GNN. Compared with CRFs, we can see that SPNs are much more efficient, because the proxy optimization problem in SPNs is much easier to solve.

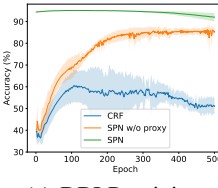
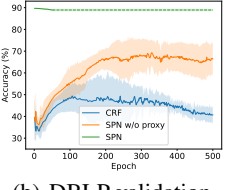
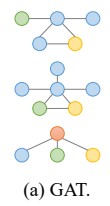
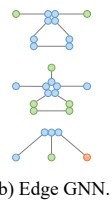
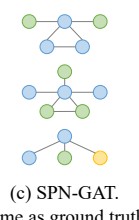

(a) DBLP training. (b) DBLP validation.

(a) GAT. (b) Edge GNN. (c) SPN-GAT. (same as ground truth)

Figure 2: Convergence curves for solving the maximin game in Eq. (3) during model learning.

Figure 3: Case study. SPN correctly predicts all node labels than GAT and the edge GNN.

**5. Comparison of learning methods.** Next, we investigate different methods for learning SPNs, including directly solving the maximin game, optimizing pseudolikelihood, and solving our proposed proxy problem. We show the results for optimizing SPN-GAT in Tab. 5. We see solving maximin game yields poor results due to unstable training. Although the pseudolikelihood method performs much better, the result is still unsatisfactory as it is not a good approximation of the true likelihood. By solving our proposed proxy problem, SPN achieves the best result, which proves its effectiveness.

**6. Convergence analysis.** To better illustrate the advantage of the proxy problem for learning CRFs, we look into the training curves of SPNs, SPNs w/o proxy, and CRFs when optimizing the maximin game in Eq. (3). For SPNs, we optimize the node and edge GNNs on the proxy optimization problem in Eq. (9) before doing refinement with the maximin game, while for SPNs w/o proxy we directly perform refinement with the maximin game without solving the proxy problem. We show the results in Fig. 2. CRFs and SPNs w/o proxy suffer from high variance and low accuracy. In contrast, owing to the near-optimal joint distribution found by solving the proxy problem, SPNs get much higher accuracy with lower variance even without refinement (see initial results of SPNs at epoch 0). Also, the refinement process quickly converges after only a few epochs, showing good efficiency of SPNs.

**7. Case study.** To intuitively see how SPNs outperform GNNs, we conduct some case studies on Cora*. We use GAT as backbone networks, and show the prediction made by the GAT (the node GNN), the edge GNN, and SPN in Fig. 3. In all three cases shown in the figure, GAT (left column) makes inconsistent predictions on linked nodes, as it fails to model the structured output. The edge GNN (middle column) also makes a mistake in the bottom case. Finally, by combining GAT and edge GNN with a CRF, the SPN (right column) is able to predict the correct labels for all nodes.

# 6 CONCLUSION

This paper studied inductive node classification, and we proposed SPN to combine GNNs and CRFs. Inspired by the connection of joint and marginal distributions defined by Markov networks, we designed a proxy problem for efficient model learning. In the future, we plan to explore more advanced GNNs to model the pseudomarginals on edges, which are key to improving node classification results in SPNs. In addition, SPNs model joint dependency of node labels by defining potential functions on nodes and edges, and we also plan to further explore high-order local structures, e.g., triangles.

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

# A    DERIVATION OF THE MAXIMIN GAME

As discussed in the Sec. 3, optimizing the joint label distribution $p_\theta$ to maximize the log-likelihood $\log p_\theta(\mathbf{y}_V^* | \mathbf{x}_V, E)$ on a training graph $(\mathbf{y}_V^*, \mathbf{x}_V, E)$ is equivalent to solving a maximin game. In this section, we provide the detailed derivation.

Let $\psi_\theta(\mathbf{y}_V, \mathbf{x}_V, E)$ be the potential function as below:

$$\psi_\theta(\mathbf{y}_V, \mathbf{x}_V, E) = \exp\left\{ \sum_{s \in V} \theta_s(y_s, \mathbf{x}_V, E) + \sum_{(s,t) \in E} \theta_{st}(y_s, y_t, \mathbf{x}_V, E) \right\}. \tag{14}$$

For each training graph $(\mathbf{y}_V^*, \mathbf{x}_V, E)$, we aim at maximizing the following log-likelihood function:

$$\begin{aligned}
\log p_\theta(\mathbf{y}_V^* | \mathbf{x}_V, E) &= \log \frac{1}{Z_\theta(\mathbf{x}_V, E)} \psi_\theta(\mathbf{y}_V^*, \mathbf{x}_V, E) \\
&= \log \psi_\theta(\mathbf{y}_V^*, \mathbf{x}_V, E) - \log Z_\theta(\mathbf{x}_V, E) \\
&= \log \psi_\theta(\mathbf{y}_V^*, \mathbf{x}_V, E) - \log \sum_{\mathbf{y}_V} \psi_\theta(\mathbf{y}_V, \mathbf{x}_V, E).
\end{aligned} \tag{15}$$

However, the term $\log \sum_{\mathbf{y}_V} \psi_\theta(\mathbf{y}_V, \mathbf{x}_V, E)$ is computationally intractable, as we need to sum over all the possible $\mathbf{y}_V$. To solve the problem, we introduce a variational joint distribution $q(\mathbf{y}_V)$ defined on all node labels $\mathbf{y}_V$, and use the Jensen's inequality to derive an estimation of the term $\log \sum_{\mathbf{y}_V} \psi_\theta(\mathbf{y}_V, \mathbf{x}_V, E)$ as follows:

$$\begin{aligned}
\log \sum_{\mathbf{y}_V} \psi_\theta(\mathbf{y}_V, \mathbf{x}_V, E) &= \log \mathbb{E}_{q(\mathbf{y}_V)} \left[ \frac{\psi_\theta(\mathbf{y}_V, \mathbf{x}_V, E)}{q(\mathbf{y}_V)} \right] \\
&\geq \mathbb{E}_{q(\mathbf{y}_V)} \left[ \log \frac{\psi_\theta(\mathbf{y}_V, \mathbf{x}_V, E)}{q(\mathbf{y}_V)} \right] \\
&= \mathbb{E}_{q(\mathbf{y}_V)}[\log \psi_\theta(\mathbf{y}_V, \mathbf{x}_V, E)] - \mathbb{E}_{q(\mathbf{y}_V)}[\log q(\mathbf{y}_V)].
\end{aligned} \tag{16}$$

The equation holds if and only if $q(\mathbf{y}_V) = p_\theta(\mathbf{y}_V | \mathbf{x}_V, E)$, and hence:

$$\log \sum_{\mathbf{y}_V} \psi_\theta(\mathbf{y}_V, \mathbf{x}_V, E) = \max_{q(\mathbf{y}_V)} \left\{ \mathbb{E}_{q(\mathbf{y}_V)}[\log \psi_\theta(\mathbf{y}_V, \mathbf{x}_V, E)] - \mathbb{E}_{q(\mathbf{y}_V)}[\log q(\mathbf{y}_V)] \right\}. \tag{17}$$

By taking the above result into Eq. (15), we obtain:

$$\begin{aligned}
\log p_\theta(\mathbf{y}_V^* | \mathbf{x}_V, E) &= \log \psi_\theta(\mathbf{y}_V^*, \mathbf{x}_V, E) - \log \sum_{\mathbf{y}_V} \psi_\theta(\mathbf{y}_V, \mathbf{x}_V, E) \\
&= \min_{q(\mathbf{y}_V)} \left\{ \log \psi_\theta(\mathbf{y}_V^*, \mathbf{x}_V, E) - \mathbb{E}_{q(\mathbf{y}_V)}[\log \psi_\theta(\mathbf{y}_V, \mathbf{x}_V, E)] + \mathbb{E}_{q(\mathbf{y}_V)}[\log q(\mathbf{y}_V)] \right\}.
\end{aligned} \tag{18}$$

As $\psi_\theta(\mathbf{y}_V, \mathbf{x}_V, E) = \exp\{ \sum_{s \in V} \theta_s(y_s, \mathbf{x}_V, E) + \sum_{(s,t) \in E} \theta_{st}(y_s, y_t, \mathbf{x}_V, E) \}$, we have:

$$\log p_\theta(\mathbf{y}_V^* | \mathbf{x}_V, E) = \min_q \mathcal{L}(\theta, q), \tag{19}$$

with:

$$\begin{aligned}
\mathcal{L}(\theta, q) = &-H[q(\mathbf{y}_V)] \\
&+ \sum_{(s,t) \in E} \{ \theta_{st}(y_s^*, y_t^*) - \mathbb{E}_{q_{st}(y_s, y_t)}[\theta_{st}(y_s, y_t)] \} + \sum_{s \in V} \{ \theta_s(y_s^*) - \mathbb{E}_{q_s(y_s)}[\theta_s(y_s)] \}.
\end{aligned} \tag{20}$$

Therefore, optimizing $\theta$ to maximize the log-likelihood function is equivalent to solving the following maximin game:

$$\max_\theta \log p_\theta(\mathbf{y}_V^* | \mathbf{x}_V, E) = \max_\theta \min_q \mathcal{L}(\theta, q). \tag{21}$$

## B    DERIVATION OF THE MOMENT-MATCHING CONDITIONS

In the CRF model defined in the preliminary section, the parameter $\theta$ consists of the output values of all $\theta$-functions. In other words, $\theta = \{\theta_s(y_s)\}_{y_s \in \mathcal{Y}, s \in V} \cup \{\theta_{st}(y_s, y_t)\}_{y_s \in \mathcal{Y}, y_t \in \mathcal{Y}, s \in V}$, where $\mathcal{Y}$ is the set of all the possible node labels.

By definition, $p_\theta(\mathbf{y}_V | \mathbf{x}_V, E)$ belongs to the exponential family. According to properties of exponential family distributions, $\log p_\theta(\mathbf{y}_V^* | \mathbf{x}_V, E)$ is strictly concave with respect to $\theta$. Therefore, the optimal $\theta$ is unique, which is characterized by the condition of $\frac{\partial}{\partial \theta} \log p_\theta(\mathbf{y}_V^* | \mathbf{x}_V, E) = 0$. Formally, $\frac{\partial}{\partial \theta} \log p_\theta(\mathbf{y}_V^* | \mathbf{x}_V, E)$ can be computed as below:

$$\frac{\partial}{\partial \theta_s(\hat{y}_s)} \log p_\theta(\mathbf{y}_V^* | \mathbf{x}_V, E) = \frac{\partial}{\partial \theta} \log \psi_\theta(\mathbf{y}_V^*, \mathbf{x}_V, E) - \frac{\partial}{\partial \theta} \log Z_\theta(\mathbf{x}_V, E). \tag{22}$$

For $\frac{\partial}{\partial \theta} \log Z_\theta(\mathbf{x}_V, E)$, we have:

$$\begin{aligned}
\frac{\partial}{\partial \theta} \log Z_\theta(\mathbf{x}_V, E) &= \frac{\partial}{\partial \theta} \log \sum_{\mathbf{y}_V} \psi_\theta(\mathbf{y}_V, \mathbf{x}_V, E) \\
&= \frac{\sum_{\mathbf{y}_V} \frac{\partial}{\partial \theta} \psi_\theta(\mathbf{y}_V, \mathbf{x}_V, E)}{\sum_{\mathbf{y}_V} \psi_\theta(\mathbf{y}_V, \mathbf{x}_V, E)} \\
&= \frac{\sum_{\mathbf{y}_V} \psi_\theta(\mathbf{y}_V, \mathbf{x}_V, E) \frac{\partial}{\partial \theta} \log \psi_\theta(\mathbf{y}_V, \mathbf{x}_V, E)}{\sum_{\mathbf{y}_V} \psi_\theta(\mathbf{y}_V, \mathbf{x}_V, E)} \\
&= \sum_{\mathbf{y}_V} \left[ \frac{\psi_\theta(\mathbf{y}_V, \mathbf{x}_V, E)}{\sum_{\mathbf{y}_V'} \psi_\theta(\mathbf{y}_V', \mathbf{x}_V, E)} \frac{\partial}{\partial \theta} \log \psi_\theta(\mathbf{y}_V, \mathbf{x}_V, E) \right] \\
&= \sum_{\mathbf{y}_V} \left[ \frac{\psi_\theta(\mathbf{y}_V, \mathbf{x}_V, E)}{Z_\theta} \frac{\partial}{\partial \theta} \log \psi_\theta(\mathbf{y}_V, \mathbf{x}_V, E) \right] \\
&= \mathbb{E}_{p_\theta(\mathbf{y}_V | \mathbf{x}_V, E)} \left[ \frac{\partial}{\partial \theta} \log \psi_\theta(\mathbf{y}_V, \mathbf{x}_V, E) \right].
\end{aligned} \tag{23}$$

By combining the above two equations, we have:

$$\frac{\partial}{\partial \theta} \log p_\theta(\mathbf{y}_V^* | \mathbf{x}_V, E) = \frac{\partial}{\partial \theta} \log \psi_\theta(\mathbf{y}_V^*, \mathbf{x}_V, E) - \mathbb{E}_{p_\theta(\mathbf{y}_V | \mathbf{x}_V, E)} \left[ \frac{\partial}{\partial \theta} \log \psi_\theta(\mathbf{y}_V, \mathbf{x}_V, E) \right]. \tag{24}$$

The potential function $\psi_\theta$ above is defined as $\psi_\theta(\mathbf{y}_V, \mathbf{x}_V, E) = \exp\{\sum_{s \in V} \theta_s(y_s, \mathbf{x}_V, E) + \sum_{(s,t) \in E} \theta_{st}(y_s, y_t, \mathbf{x}_V, E)\}$. If we consider each specific scalar $\theta_s(\hat{y}_s)$, and taking the derivative with respect to the scalar to 0, we obtain:

$$\begin{aligned}
0 &= \frac{\partial}{\partial \theta_s(\hat{y}_s)} \log p_\theta(\mathbf{y}_V^* | \mathbf{x}_V, E) \\
&= \frac{\partial}{\partial \theta_s(\hat{y}_s)} \log \psi_\theta(\mathbf{y}_V^*, \mathbf{x}_V, E) - \mathbb{E}_{p_\theta(\mathbf{y}_V | \mathbf{x}_V, E)} \left[ \frac{\partial}{\partial \theta_s(\hat{y}_s)} \log \psi_\theta(\mathbf{y}_V, \mathbf{x}_V, E) \right] \\
&= \mathbb{I}_{y_s^*}\{\hat{y}_s\} \left[ \frac{\partial}{\partial \theta_s(\hat{y}_s)} \theta_s(\hat{y}_s) \right] - \sum_{\mathbf{y}_V} p_\theta(\mathbf{y}_V | \mathbf{x}_V, E) \left[ \mathbb{I}_{y_s^*}\{\hat{y}_s\} \frac{\partial}{\partial \theta_s(\hat{y}_s)} \theta_s(\hat{y}_s) \right] \\
&= \mathbb{I}_{y_s^*}\{\hat{y}_s\} \left[ \frac{\partial}{\partial \theta_s(\hat{y}_s)} \theta_s(\hat{y}_s) \right] - p_\theta(\hat{y}_s | \mathbf{x}_V, E) \left[ \frac{\partial}{\partial \theta_s(\hat{y}_s)} \theta_s(\hat{y}_s) \right] \\
&= \mathbb{I}_{y_s^*}\{\hat{y}_s\} - p_\theta(\hat{y}_s | \mathbf{x}_V, E),
\end{aligned} \tag{25}$$

which implies $p_\theta(\hat{y}_s | \mathbf{x}_V, E) = \mathbb{I}_{y_s^*}\{\hat{y}_s\}$ for the optimal $\theta$. Moreover, this equation holds for all $s \in V$ and all $\hat{y}_s \in \mathcal{Y}$.

Similarly, for each scalar $\theta_{st}(\hat{y}_s, \hat{y}_t)$, we have that $\frac{\partial}{\partial \theta_{st}(\hat{y}_s, \hat{y}_t)} \log p_\theta(\mathbf{y}_V^* | \mathbf{x}_V, E) = 0$ is equivalent to $p_\theta(\hat{y}_s, \hat{y}_t | \mathbf{x}_V, E) = \mathbb{I}_{y_s^*, y_t^*}\{\hat{y}_s, \hat{y}_t\}$. This equation holds for all $(s, t) \in E$ and all the choices of $(\hat{y}_s, \hat{y}_t) \in \mathcal{Y} \times \mathcal{Y}$.

Therefore, the optimal $\theta$-functions are characterized by the moment-matching conditions as below:

$$p_\theta(y_s | \mathbf{x}_V, E) = \mathbb{I}_{y_s^*}\{y_s\} \ \forall s \in V, \quad p_\theta(y_s, y_t | \mathbf{x}_V, E) = \mathbb{I}_{y_s^*, y_t^*}\{y_s, y_t\} \ \forall (s, t) \in E. \tag{26}$$

## C  PROOF OF PROPOSITION 1

Next, we prove Prop. 1. We first restate the proposition as follows:

**Proposition** *Consider a set of nonzero pseudomarginals $\{\tau_s(y_s)\}_{s \in V}$ and $\{\tau_{st}(y_s, y_t)\}_{(st) \in E}$ which satisfy $\sum_{y_s} \tau_{st}(y_s, y_t) = \tau_t(y_t)$ and $\sum_{y_t} \tau_{st}(y_s, y_t) = \tau_s(y_s)$ for all $(s, t) \in E$.*
*If we parameterize the $\theta$-functions of $p_\theta$ in Eq. (2) in the following way:*

$$\theta_s(y_s) = \log \tau_s(y_s) \quad \forall s \in V, \quad \theta_{st}(y_s, y_t) = \log \frac{\tau_{st}(y_s, y_t)}{\tau_s(y_s)\tau_t(y_t)} \quad \forall (s, t) \in E, \qquad (27)$$

*then $\{\tau_s(y_s)\}_{s \in V}$ and $\{\tau_{st}(y_s, y_t)\}_{(s,t) \in E}$ are specified by a fixed point of the sum-product loopy belief propagation algorithm when applied to the joint distribution $p_\theta$, which implies that:*

$$\tau_s(y_s) \approx p_\theta(y_s) \quad \forall s \in V, \quad \tau_{st}(y_s, y_t) \approx p_\theta(y_s, y_t) \quad \forall (s, t) \in E. \qquad (28)$$

**Proof:** To prove the proposition, we first summarize the workflow of the sum-product loopy belief propagation algorithm. In sum-product loopy belief propagation, we introduce a message function $m_{t \to s}(y_s)$ for each edge $(s, t) \in E$. Then we iteratively update all message functions as follows:

$$m_{t \to s}(y_s) \propto \sum_{y_t} \left\{ \exp(\theta_t(y_t) + \theta_{st}(y_s, y_t)) \prod_{s' \in N(t) \backslash s} m_{s' \to t}(y_t) \right\}, \qquad (29)$$

where $N(t)$ represents the set of neighbors for node $t$.

Once the process converges or after sufficient iterations, the approximation of the node marginals and the edge marginals (i.e., $\{q_s(y_s)\}_{s \in V}$ and $\{q_{st}(y_s, y_t)\}_{(s,t) \in E}$) can be recovered by the message functions $\{m_{t \to s}(y_s)\}_{(s,t) \in E}$ as follows:

$$q_s(y_s) \propto \exp(\theta_s(y_s)) \prod_{t \in N(s)} m_{t \to s}(y_s), \qquad (30)$$

$$q_{st}(y_s, y_t) \propto \exp(\theta_s(y_s) + \theta_t(y_t) + \theta_{st}(y_s, y_t)) \prod_{t' \in N(s) \backslash t} m_{t' \to s}(y_s) \prod_{s' \in N(t) \backslash s} m_{s' \to t}(y_t). \qquad (31)$$

Next, let us move back to our case, where we parameterize the $\theta$-functions with a set of pseudo-marginals as in Eq. (27). For such a specific parameterization of the $\theta$-functions, we claim that one fixed point of Eq. (29) is achieved when $m_{t \to s}(y_s) = 1$ for all $(s, t) \in E$. To prove that, we notice that when all the message functions equal to 1, the left side of Eq. (29) is apparently 1. The right side of Eq. (29) can be computed as below:

$$\begin{aligned}
& \sum_{y_t} \exp(\theta_t(y_t) + \theta_{st}(y_s, y_t)) \prod_{s' \in N(t) \backslash s} m_{s' \to t}(y_t) \\
= & \sum_{y_t} \exp(\theta_t(y_t) + \theta_{st}(y_s, y_t)) \\
= & \sum_{y_t} \exp\left( \log \tau_t(y_t) + \log \frac{\tau_{st}(y_s, y_t)}{\tau_s(y_s)\tau_t(y_t)} \right) \\
= & \sum_{y_t} \exp\left( \log \frac{\tau_{st}(y_s, y_t)}{\tau_s(y_s)} \right) \qquad (32) \\
= & \sum_{y_t} \frac{\tau_{st}(y_s, y_t)}{\tau_s(y_s)} \\
= & \frac{\tau_s(y_s)}{\tau_s(y_s)} \\
= & 1.
\end{aligned}$$

We can see that both the left side and the right side of Eq. (29) are 1, and hence $\{m_{t \to s}(y_s) = 1\}_{(s,t) \in E}$ specifies a fixed point of sum-product loopy belief propagation. For this fixed point, $q_s(y_s)$ can be computed as follows:

$$q_s(y_s) \propto \exp(\theta_s(y_s)) \prod_{t \in N(s)} m_{t \to s}(y_s) = \exp(\theta_s(y_s)) = \tau_s(y_s). \qquad (33)$$

Similarly, we can compute $q_{st}(y_s, y_t)$ as:

$$
\begin{aligned}
q_{st}(y_s, y_t) &\propto \exp(\theta_s(y_s) + \theta_t(y_t) + \theta_{st}(y_s, y_t)) \prod_{t' \in N(s) \backslash t} m_{t' \to s}(y_s) \prod_{s' \in N(t) \backslash s} m_{s' \to t}(y_t) \\
&= \exp(\theta_s(y_s) + \theta_t(y_t) + \theta_{st}(y_s, y_t)) \\
&= \exp \left( \log \tau_s(y_s) + \log \tau_t(y_t) + \log \frac{\tau_{st}(y_s, y_t)}{\tau_s(y_s)\tau_t(y_t)} \right) \\
&= \tau_{st}(y_s, y_t).
\end{aligned}
\tag{34}
$$

From the above two equations, we can see that $\{\tau_s(y_s)\}_{s \in V}$ and $\{\tau_{st}(y_s, y_t)\}_{(s,t) \in E}$ are specified by a fixed point (i.e., $m_{t \to s}(y_s) = 1$ for all $(s,t) \in E$) of sum-product loopy belief propagation. As sum-product loopy belief propagation often works well in practice to approximate the marginal distributions on nodes and edges, we thus have $\tau_s(y_s) \approx p_\theta(y_s)$ for each node and $\tau_{st}(y_s, y_t) \approx p_\theta(y_s, y_t)$ for each edge.

# D    SOLVING THE PROXY PROBLEM WITH CONSTRAINED OPTIMIZATION

The key innovation of our proposed approach is on the proxy optimization problem which is used to approximate the original learning problem. Formally, the proxy optimization problem is stated as:

$$
\begin{aligned}
\min_{\tau, \theta} &\sum_{s \in V} d\left( \mathbb{I}_{y_s^*}\{y_s\}, \tau_s(y_s) \right) + \sum_{(s,t) \in E} d\left( \mathbb{I}_{(y_s^*, y_t^*)}\{(y_s, y_t)\}, \tau_{st}(y_s, y_t) \right), \\
\text{subject to} \quad &\theta_s = \log \tau_s(y_s), \quad \theta_{st}(y_s, y_t) = \log \frac{\tau_{st}(y_s, y_t)}{\tau_s(y_s)\tau_t(y_t)}, \\
\text{and} \quad &\sum_{y_s} \tau_{st}(y_s, y_t) = \tau_t(y_t), \quad \sum_{y_t} \tau_{st}(y_s, y_t) = \tau_s(y_s),
\end{aligned}
\tag{35}
$$

for all nodes and edges, where $d$ can be any divergence measure between two distributions.

In our implementation, we ignore these consistency constraints, i.e., $\sum_{y_s} \tau_{st}(y_s, y_t) = \tau_t(y_t)$ and $\sum_{y_t} \tau_{st}(y_s, y_t) = \tau_s(y_s)$. This ie because by by optimizing the objective, the obtained pseudo-marginals $\tau$ would well approximate the observed node and edge marginals, i.e., $\tau_s(y_s) \approx \mathbb{I}_{y_s^*}\{y_s\}$ and $\tau_{st}(y_s, y_t) \approx \mathbb{I}_{(y_s^*, y_t^*)}\{(y_s, y_t)\}$, and hence $\tau$ would almost naturally satisfy the constraints.

To demonstrate ignoring the consistency constraint makes sense, we also tried a constrained optimization method for solving the proxy problem. Specifically, we add a quadratic term to penalize the inconsistency between $\tau_{st}(y_s, y_t)$ and $\tau_s(y_s)$ as well as $\tau_t(y_t)$, resulting in the following problem:

$$
\begin{aligned}
\min_{\tau, \theta} &\sum_{s \in V} d\left( \mathbb{I}_{y_s^*}\{y_s\}, \tau_s(y_s) \right) + \sum_{(s,t) \in E} d\left( \mathbb{I}_{(y_s^*, y_t^*)}\{(y_s, y_t)\}, \tau_{st}(y_s, y_t) \right) \\
&+ \alpha \sum_{(s,t) \in E} \left[ \sum_{y_t} \{ \sum_{y_s} \tau_{st}(y_s, y_t) - \tau_t(y_t) \}^2 + \sum_{y_s} \{ \sum_{y_t} \tau_{st}(y_s, y_t) - \tau_s(y_s) \}^2 \right], \\
\text{subject to} \quad &\theta_s = \log \tau_s(y_s), \quad \theta_{st}(y_s, y_t) = \log \frac{\tau_{st}(y_s, y_t)}{\tau_s(y_s)\tau_t(y_t)},
\end{aligned}
\tag{36}
$$

for all nodes and edges. Again, $d$ is a divergence measure between two distributions, and we choose to use the KL divergence. $\alpha$ is a hyperparameter deciding the weight of the penalty term.

Table 7: Analysis of constrained optimization methods for solving the proxy problem.

| Algorithm | Constrained Optimization | Cora* | Citeseer* | Pubmed* |
|---|---|---|---|---|
| SPN-GAT | w/o | **49.10** ± 3.80 | **42.89** ± 1.30 | **47.79** ± 1.33 |
|  | with | 48.83 ± 3.51 | 42.04 ± 1.23 | 47.55 ± 1.24 |

We conduct empirical comparison of this constrained optimization method and our default implementation where the consistency constraint is ignored. The results are presented in Tab. 7. We can see that the constrained optimization method does not lead to improvement, which shows that ignoring the consistency constraint is empirically reasonable.

# E UNDERSTANDING SPNS AS OPTIMIZING A SURROGATE FOR THE LOG-LIKELIHOOD FUNCTION

In the model section, we motivate SPNs from the moment-matching conditions of the optimal $\theta$-functions. Specifically, we initialize the $\theta$-functions at a state where the moment-matching conditions are approximately satisfied, yielding a near-optimal joint distribution. Then we further tune the $\theta$-functions to solve the maximin game. Besides this perspective, SPNs can also be understood as optimizing a surrogate for the log-likelihood function. Next, we introduce the details.

Remember that maximizing the log-likelihood function is equivalent to solving a maximin game as:

$$\max_{\theta} \log p_{\theta}(\mathbf{y}_V^* | \mathbf{x}_V, E) = \max_{\theta} \min_{q} \mathcal{L}(\theta, q), \quad \text{with} \quad \mathcal{L}(\theta, q) = -H[q(\mathbf{y}_V)]$$
$$+ \sum_{s \in V} \{\theta_s(y_s^*) - \mathbb{E}_{q_s(y_s)}[\theta_s(y_s)]\} + \sum_{(s,t) \in E} \{\theta_{st}(y_s^*, y_t^*) - \mathbb{E}_{q_{st}(y_s, y_t)}[\theta_{st}(y_s, y_t)]\}. \tag{37}$$

Here, $q(\mathbf{y}_V)$ is a joint distribution on all the node labels. $q_s(y_s)$ and $q_{st}(y_s, y_t)$ are the corresponding marginal distributions.

Although the above maximin game is equivalent to the original problem of maximizing likelihood, solving the maximin game is nontrivial. In particular, there are two key challenges, i.e., (1) how to specify constraints to characterize a valid joint distribution $q(\mathbf{y}_V)$ and (2) how to compute its entropy $H(q) = -\mathbb{E}_{q(\mathbf{y}_V)}[\log q(\mathbf{y}_V)]$. To deal with the challenge, a common practice used in loopy belief propagation is to make the following two approximations:

(1) Instead of specifying constraints to let $q(\mathbf{y}_V)$ be a valid joint distribution, we introduce a set of pseudomarginals as approximation to a valid joint distribution. Specifically, these pseudomarginals are denoted as $\tilde{q} = \{q_s(y_s)\}_{s \in V} \cup \{q_{st}(y_s, y_t)\}_{(s,t) \in E}$, and they satisfy $\sum_{y_s} q_{st}(y_s, y_t) = q_t(y_t)$ and $\sum_{y_t} q_{st}(y_s, y_t) = q_s(y_s)$ for all $(s, t) \in E$.

(2) We approximate the entropy $H(q)$ with Bethe entropy approximation $H_{\text{Bethe}}(\tilde{q})$, which is defined as follows:

$$H_{\text{Bethe}}(\tilde{q}) = -\sum_{s \in V} \mathbb{E}_{q_s(y_s)}[\log q_s(y_s)] - \sum_{(s,t) \in E} \mathbb{E}_{q_{st}(y_s, y_t)} \left[ \log \frac{q_{st}(y_s, y_t)}{q_s(y_s) q_t(y_t)} \right]. \tag{38}$$

With the two approximations, we get the following maximin game as a surrogate for the likelihood maximization problem:

$$\max_{\theta} \log p_{\theta}(\mathbf{y}_V^* | \mathbf{x}_V, E) \approx \max_{\theta} \min_{\tilde{q}} \mathcal{L}_{\text{Bethe}}(\theta, \tilde{q}), \tag{39}$$

with:

$$\mathcal{L}_{\text{Bethe}}(\theta, \tilde{q}) = -H_{\text{Bethe}}(\tilde{q})$$
$$+ \sum_{(s,t) \in E} \{\theta_{s,t}(y_s^*, y_t^*) - \mathbb{E}_{q_{st}(y_s, y_t)}[\theta_{s,t}(y_s, y_t)]\} + \sum_{s \in V} \{\theta_s(y_s^*) - \mathbb{E}_{q_s(y_s)}[\theta_s(y_s)]\}. \tag{40}$$

This problem is known as the Bethe variational problem (BVP) (Wainwright & Jordan, 2008).

Such a problem can be solved by coordinate descent, where we alternate between updating $\tilde{q}$ to minimize $\mathcal{L}_{\text{Bethe}}(\theta, \tilde{q})$ and updating $\theta$ to maximize $\mathcal{L}_{\text{Bethe}}(\theta, \tilde{q})$. According to Yedidia et al. (2005), updating $\tilde{q}$ to minimize $\mathcal{L}_{\text{Bethe}}(\theta, \tilde{q})$ can be exactly achieved by running sum-product loopy belief propagation on $p_{\theta}$, where a fixed point of the belief propagation algorithm yields a local optima of $\tilde{q}$. On the other hand, updating $\theta$ to maximize $\mathcal{L}_{\text{Bethe}}(\theta, \tilde{q})$ can be easily achieved by gradient ascent.

In addition to that, a stationary point $(\theta^*, \tilde{q}^*)$ of the above BVP is specified by following conditions:

$$\frac{\partial \mathcal{L}_{\text{Bethe}}(\theta^*, \tilde{q}^*)}{\partial \tilde{q}^*} = 0 \qquad \frac{\partial \mathcal{L}_{\text{Bethe}}(\theta^*, \tilde{q}^*)}{\partial \theta^*} = 0. \tag{41}$$

According to Yedidia et al. (2005) and Wainwright & Jordan (2008), the first condition is equivalent to the condition that $\tilde{q}^*$ is specified by a fixed-point of sum-product loopy belief propagation. The second condition states that the moment-matching conditions are satisfied, i.e., $q_s(y_s) = \mathbb{I}_{y_s^*}\{y_s\}$ on each node and $q_{st}(y_s, y_t) = \mathbb{I}_{(y_s^*, y_t^*)}\{(y_s, y_t)\}$ on each edge.

For our proposed approach SPN, it can be viewed as solving the BVP as defined in Eq. (39). Through solving the proxy problem, SPN initializes $\theta$ at a state where the conditions of stationary points in Eq. (41) are approximately satisfied. Then the fine-tuning stage of SPN further adjusts $\theta$ to solve the maximin game by alternatively updating $\theta$ and $\tilde{q}$.

More specifically, when solving the proxy optimization problem, by initializing $\theta$ in the way defined by Eq. (27), the collection of pseudomarginal distributions $\{\tau_s(y_s)\}_{s \in V}$ and $\{\tau_{st}(y_s, y_t)\}_{(s,t) \in E}$ is specified by a fixed point of sum-product loopy belief propagation according to Prop. 1. This implies that $\frac{\partial}{\partial \tilde{q}} \mathcal{L}_{\text{Bethe}}(\theta, \tilde{q}) = 0$ for $\tilde{q} = \{\tau_s(y_s)\}_{s \in V} \cup \{\tau_{st}(y_s, y_t)\}_{(s,t) \in E}$. Meanwhile, as $\{\tau_s\}_{s \in V}$ and $\{\tau_{st}\}_{(s,t) \in E}$ are learned to match the true labels $\mathbf{y}_V^*$ on each training graph, we thus have $\tau_s(y_s) \approx \mathbb{I}_{y_s^*}\{y_s\}$ on each node and $\tau_{st}(y_s, y_t) \approx \mathbb{I}_{(y_s^*, y_t^*)}\{(y_s, y_t)\}$ on each edge. Therefore, the conditions in Eq. (41) are approximately satisfied by $(\theta, \tilde{q})$ with $\tilde{q} = \{\tau_s(y_s)\}_{s \in V} \cup \{\tau_{st}(y_s, y_t)\}_{(s,t) \in E}$, which means that solving the proxy problem yields a $\theta$ to roughly match the conditions of stationary points for the BVP in Eq. (39). Afterwards, the refinement stage of SPN is exactly trying to solve the maximin game of BVP in Eq. (39), where we alternate between updating $\tilde{q}$ to minimize $\mathcal{L}_{\text{Bethe}}(\theta, \tilde{q})$ via sum-product loopy belief propagation and updating $\theta$ to maximize $\mathcal{L}_{\text{Bethe}}(\theta, \tilde{q})$ via gradient ascent.

As a result, we see that the SPN can also be understood as solving the Bethe variational problem in Eq. (39), which acts as a surrogate for the log-likelihood function.

## F  EXPERIMENTAL DETAILS

Next, we describe our experimental setup in more details.

### F.1  DATASETS

The statistics of the datasets used in our experiment are summarized in Tab. 8. For the Cora*, Citeseer*, Pubmed*, and PPI datasets, they are under the MIT license.

Table 8: Dataset statistics. ML and MC stand for multi-label classification and multi-class classification respectively.

| Dataset | Task | # Features | # Labels | Training Graphs | | | Validation Graphs | | | Test Graphs | | |
|---|---|---|---|---|---|---|---|---|---|---|---|---|
| | | | | # Graphs | Avg. # Nodes | Avg. # Edges | # Graphs | Avg. # Nodes | Avg. # Edges | # Graphs | Avg. # Nodes | Avg. # Edges |
| PPI | ML | 50 | 121 | 20 | 2245.3 | 61318.4 | 2 | 3257 | 99460.0 | 2 | 2762 | 80988.0 |
| Cora* | MC | 1433 | 7 | 140 | 5.6 | 7.0 | 500 | 4.9 | 5.8 | 1000 | 4.7 | 5.3 |
| Citeseer* | MC | 3703 | 6 | 120 | 4.0 | 4.3 | 500 | 3.8 | 4.0 | 1000 | 3.8 | 3.8 |
| Pubmed* | MC | 500 | 3 | 60 | 6.0 | 6.7 | 500 | 5.4 | 5.8 | 1000 | 5.6 | 6.7 |
| DBLP | MC | 100 | 3 | 1 | 6488 | 10262 | 1 | 14142 | 48631 | 1 | 26813 | 155899 |

For the DBLP dataset, it is constructed from the citation network [4] in Tang et al. (2008). Scientific papers from eight conferences are treated as nodes, which are divided into three categories based on conference domains [5] for classification. For each paper, we compute the mean GloVe embedding [6] (Pennington et al., 2014) of words in the title and abstract as features. We split the dataset into three disjoint graphs for training/validation/test. The training graph contains papers published before 1999 (with 1999 included). The validation graph contains papers published between 2000 and 2009 (with 2000 and 2009 included). The test graph contains papers published after 2010 (with 2010 included). There exists an undirected edge between two papers if one cites the other one. Cross-split edges (e.g., an edge between a paper in the training set and a paper in the validation set) are removed.

For the PPI datasets, there are 121 binary labels, and we treat each binary label as an independent task. For each compared algorithm, we train a separate model for each task, and report the overall results across all tasks.

### F.2  ARCHITECTURE CHOICES

To facilitate reproducibility, we use the GNN module implementations of PyTorch Geometric (Fey & Lenssen, 2019), and follow the GNN models provided in the examples of the repository, unless

---

[4] https://originalstatic.aminer.cn/misc/dblp.v12.7z
[5] ML: ICML/NeurIPS. CV: ICCV/CVPR/ECCV. NLP: ACL/EMNLP/NAACL.
[6] http://nlp.stanford.edu/data/glove.6B.zip

otherwise mentioned. Note that most architecture choices are not optimal on the benchmark datasets, but we did not tune them since we only aim to show that our method brings consistent and significant improvement.

**GCN (Kipf & Welling, 2017).** We set the number of hidden neurons to 16, and the number of layers to 2. ReLU (Nair & Hinton, 2010) is used as the activation function. We do not dropout between GNN layers.

**GraphSage (Hamilton et al., 2017).** We set the number of hidden neurons to 64, and the number of layers to 2. ReLU (Nair & Hinton, 2010) is used as the activation function. We do not dropout between GNN layers.

**GAT (Veličković et al., 2018).** We set the number of hidden neurons to 256 per attention head, and the number of layers to 3. The number of heads for each layer is set to 4, 4 and 6. ELU (Clevert et al., 2016) is used as the activation function. We do not dropout between GNN layers.

**Graph U-Net (Gao & Ji, 2019).** We set the number of hidden neurons to 64 and the number of layers to 3. We randomly dropout 20% of the edges from the adjacency matrix. We do not dropout node features or between layers.

**GCNII (Chen et al., 2020a).** We set the number of hidden neurons to 2048 for the citation datasets (Cora*, Citeseer*, Pubmed* and DBLP) and 256 for the PPI dataset. We set the number of layers to 9. ReLU (Nair & Hinton, 2010) is used as the activation function. For PPI, layer normalization (Ba et al., 2016) is applied between the GCNII layers. We do not dropout between GNN layers. We set the strength $\alpha$ of the initial residual connection to 0.5, and the hyperparameter $\theta$ to compute the strength of the identity mapping to 1.

**The $g$ function.** In Eq. (8) of the model section, we define $g$ as a function mapping a pair of $L$-dimensional representations to a $(|\mathcal{Y}| \times |\mathcal{Y}|)$-dimensional logit. Two variants of this function are used in our experiment. For the PPI and DBLP dataset, we use the linear variant, where the pair of node representations are concatenated and plugged into a linear layer:

$$g_{\text{linear}}(\mathbf{v}_s, \mathbf{v}_t) = \mathbf{W}[\mathbf{v}_s; \mathbf{v}_t] + b, \tag{42}$$

where $\mathbf{W} \in \mathbb{R}^{(|\mathcal{Y}| \times |\mathcal{Y}|) \times 2L}$ is the weight matrix and $b \in \mathbb{R}^{|\mathcal{Y}| \times |\mathcal{Y}|}$ is the bias. For the citation datasets (Cora*, Citeseer*, Pubmed*), we use the bilienar variant, where the pair of node representations are plugged in a bilinear mapping:

$$g_{\text{bilinear}}(\mathbf{v}_s, \mathbf{v}_t) = (\mathbf{W}\mathbf{v}_s)(\mathbf{W}\mathbf{v}_t)^T, \tag{43}$$

where $\mathbf{W} \in \mathbb{R}^{|\mathcal{Y}| \times L}$ is a weight matrix.

**SPN with a shared GNN.** By default, the SPN uses a node GNN and an edge GNN to approximate the pseudomarginals on nodes and edges respectively. In the experiment, we also consider using a shared GNN for both pseudomarginals on nodes and edges. In other words, $\mathbf{u}_s = \mathbf{v}_s, \forall s \in V$ (see Eq. (7) and Eq. (8)). All the other components are the same as the default SPN. The results of this variant are shown in Tab. 6 of the experiment section.

### F.3 HYPERPARAMETER CHOICES

**GNNs and SPNs.** For node classification, the learning rate of the node GNN $\tau_s$ in GNNs and SPNs is presented in Tab. 9. For edge classification, the learning rate of the edge GNN $\tau_{st}$ is presented in Tab. 10. For the temperature $\gamma$ used in the edge GNN $\tau_{st}$ of SPNs, we report its values in Tab. 11.

**CRF-linear.** For CRF-linear training, we set the learning rate to $5 \times 10^{-4}$.

**CRF-GNNs and SPN.** For CRF and the refinement stage of SPN, we set learning rates to $1 \times 10^{-5}$.

**GMNN.** For GMNN training, we set the learning rate to $5 \times 10^{-3}$.

Table 9: Learning rate of the node GNN $\tau_s$.

| Algorithm | PPI | Cora* | Citeseer* | Pubmed* | DBLP |
|-----------|-----|-------|-----------|---------|------|
| GCN | $5 \times 10^{-3}$ | $5 \times 10^{-3}$ | $1 \times 10^{-2}$ | $1 \times 10^{-2}$ | $1 \times 10^{-2}$ |
| GraphSage | $5 \times 10^{-3}$ | $5 \times 10^{-3}$ | $5 \times 10^{-3}$ | $5 \times 10^{-3}$ | $5 \times 10^{-3}$ |
| GAT | $5 \times 10^{-3}$ | $1 \times 10^{-2}$ | $1 \times 10^{-3}$ | $1 \times 10^{-3}$ | $1 \times 10^{-3}$ |
| Graph U-Net | $5 \times 10^{-3}$ | $1 \times 10^{-2}$ | $1 \times 10^{-2}$ | $1 \times 10^{-2}$ | - |
| GCNII | $5 \times 10^{-3}$ | $1 \times 10^{-2}$ | $1 \times 10^{-2}$ | $1 \times 10^{-2}$ | $1 \times 10^{-3}$ |

Table 10: Learning rate of the edge GNN $\tau_{st}$.

| Algorithm | PPI | Cora* | Citeseer* | Pubmed* | DBLP |
|-----------|-----|-------|-----------|---------|------|
| GCN | $1 \times 10^{-3}$ | $1 \times 10^{-2}$ | $5 \times 10^{-2}$ | $1 \times 10^{-2}$ | $5 \times 10^{-3}$ |
| GraphSage | $1 \times 10^{-3}$ | $1 \times 10^{-3}$ | $1 \times 10^{-3}$ | $1 \times 10^{-3}$ | $1 \times 10^{-3}$ |
| GAT | $1 \times 10^{-3}$ | $1 \times 10^{-3}$ | $5 \times 10^{-4}$ | $5 \times 10^{-4}$ | $5 \times 10^{-4}$ |
| Graph U-Net | $1 \times 10^{-3}$ | $1 \times 10^{-2}$ | $1 \times 10^{-2}$ | $1 \times 10^{-2}$ | - |
| GCNII | $1 \times 10^{-3}$ | $5 \times 10^{-3}$ | $1 \times 10^{-3}$ | $1 \times 10^{-3}$ | $1 \times 10^{-4}$ |

Table 11: Temperature $\gamma$ of the edge GNN $\tau_{st}$.

| Algorithm | PPI | Cora* | Citeseer* | Pubmed* | DBLP |
|-----------|-----|-------|-----------|---------|------|
| GCN | 10 | 0.2 | 1 | 2 | 2 |
| GraphSage | 10 | 10 | 10 | 10 | 10 |
| GAT | 10 | 0.2 | 10 | 0.2 | 0.2 |
| Graph U-Net | 10 | 0.5 | 1 | 0.2 | - |
| GCNII | 10 | 0.5 | 0.5 | 0.5 | 2 |

### F.4 COMPUTATIONAL RESOURCES

We run the experiment by using NVIDIA Tesla V100 GPUs with 16GB memory.

## G ADDITIONAL RESULTS

In this section, we present some additional experimental results.

### G.1 ADDITIONAL ANALYSIS OF GNN ARCHITECTURES

In this analysis, we study the effect of node/edge GNN architectures on SPNs. We fix one of the GNNs and change the capacity of the other (Fig. 4), then evaluate SPN-GAT on **PPI-1-0**, a subset of PPI-1 that only contains its first label. The results show that our model benefit from capacity gain in both node and edge GNNs, highlighting their effective synergy. This also explains the underperformance of SPN-GCN in Tab. 1, where the edge GCN backbone with only two layers and 16 hidden neurons is incapable of modeling the edge label dependencies and thus drags the performance behind. We also find that the node and edge GNNs need not share the same backbone, and in many cases SPNs with different node and edge GNNs perform superior to those with same backbone (Fig. 4). The expressiveness of edge GNNs is crucial to the performance of SPN. Though we did not optimize the design of our edge GNNs, they have shown to be helpful in boosting the performance once plugged into our approach.

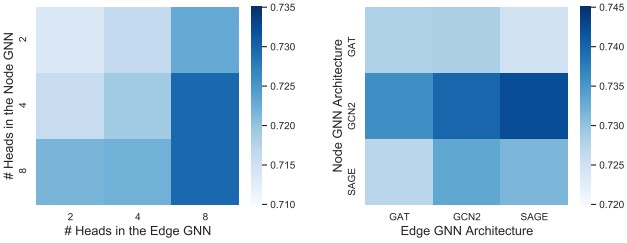

Figure 4: Left: effect of #heads in SPN-GAT. Right: effect of backbones in SPN.

## G.2    NODE-LEVEL ACCURACY ON CORA*, CITESEER*, AND PUBMED*

In the experiment, we report the graph-level accuracy on the Cora*, Citeseer*, and Pubmed* datasets, where SPNs consistently outperform other methods. Besides the graph-level accuracy, we also compute the node-level accuracy on these datasets, and the results are reported in Tab. 12. We can see that our approach still consistently outperforms other methods in terms of node-level accuracy.

Table 12: Node-level accuracy on Cora*, Citeseer*, Pubmed* (in %).

| Algorithm | Cora* | Citeseer* | Pubmed* |
|---|---|---|---|
| GCN | $79.85 \pm 0.24$ | $72.25 \pm 0.71$ | $78.05 \pm 0.55$ |
| GraphSAGE | $73.43 \pm 1.67$ | $62.48 \pm 2.19$ | $73.99 \pm 1.26$ |
| GAT | $79.65 \pm 1.25$ | $74.15 \pm 0.12$ | $78.62 \pm 0.52$ |
| Graph U-Net | $78.72 \pm 0.63$ | $71.36 \pm 1.37$ | $77.93 \pm 0.60$ |
| GCNII | $82.84 \pm 0.37$ | $72.61 \pm 0.49$ | $79.47 \pm 0.55$ |
| CRF-linear | $68.47 \pm 2.13$ | $65.88 \pm 0.85$ | $65.93 \pm 2.18$ |
| CRF-GAT | $77.75 \pm 1.24$ | $69.13 \pm 1.10$ | $75.96 \pm 1.06$ |
| CRF-UNet | $78.32 \pm 1.51$ | $70.78 \pm 1.15$ | $77.91 \pm 0.56$ |
| CRF-GCNII | $35.98 \pm 7.40$ | $33.73 \pm 5.87$ | $60.55 \pm 4.17$ |
| GMNN | $79.90 \pm 0.93$ | $72.18 \pm 0.48$ | $78.00 \pm 1.04$ |
| SPN-GAT | $83.13 \pm 0.48$ | $\mathbf{74.50} \pm 0.36$ | $79.23 \pm 0.33$ |
| SPN-UNet | $81.11 \pm 0.55$ | $72.28 \pm 0.94$ | $78.70 \pm 0.37$ |
| SPN-GCNII | $\mathbf{83.54} \pm 0.27$ | $74.04 \pm 0.29$ | $\mathbf{79.95} \pm 0.38$ |

## G.3    COMPARISON OF SUM-PRODUCT AND MAX-PRODUCT BELIEF PROPAGATION

As explained in section 4.2, the sum-product belief propagation algorithm is more applicable to the case of node-level accuracy, as it aims at inferring the marginal label distribution on each node. Nevertheless, in practice we find that the max-product algorithm usually achieves better empirical node-level accuracy. For example, the results on the PPI-10 dataset are presented in Tab. 13.

Table 13: Micro-F1 on PPI-10 (in %).

| Algorithm | Micro-F1 |
|---|---|
| Sum-product BP | $94.50 \pm 0.16$ |
| Max-product BP | $94.65 \pm 0.13$ |

Because of the better empirical results, we choose to use max-product belief propagation by default.

## G.4    HYPERPARAMETER ANALYSIS

Finally, We present analysis of the hyperparameter $\gamma$ (i.e., edge temperature) in Fig. 5.

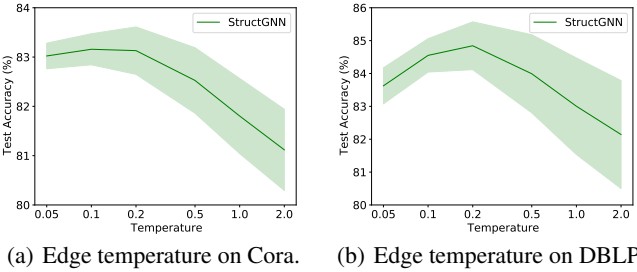

(a) Edge temperature on Cora.    (b) Edge temperature on DBLP.

Figure 5: Hyperparameter analysis.

