# OpenReview forum: "Neural Structured Prediction for Inductive Node Classification"
_ICLR.cc/2022/Conference — ICLR 2022 Oral_

### Official Review · Reviewer_jqoo · 2021-11-02

**Correctness:** 4
**Technical Novelty And Significance:** 3
**Empirical Novelty And Significance:** 2
**Recommendation:** 8
**Confidence:** 4

**Main Review:**

Strengths
- The motivation of model design is clear and reasonable. Modeling label dependencies is a natural way to improve GNNs, which is also supported by experiments.
- The model inherits advantages of CRF, such as describing the dependency of node labels and providing probabilistic interpretation. In the meantime, the model supports efficient learning and inference.
- The model is technically sound.
- Experiments cover several benchmarks. The model is tested on multiple datasets and shows very promising results.
- The paper is well-written and easy to follow.

Weaknesses
- The model is proposed for modeling discrete node labels on a graph. It is unclear how to extend the model to fit continuous node labels.
- Insufficient baselines for comparison. For neural models that can describe node label dependencies, the paper compares with GMNN only. However, there are various recent works that are not compared, e.g., $G^3NN$ (Ma et al., 2019), CopulaGNN (Ma et al., 2021), LCM (Wang et al., 2021) to cite a few.

**Summary Of The Paper:**

The paper aims to break the label independence assumption in existing GNNs. To this end, a combination of GNNs and CRF is proposed, named SMN. SMN improves GNNs by modeling the joint distribution among discrete node labels; SMN is a promising alternative to CRF as it provides efficient learning and inference procedures. Empirically, SMN achieves better node-level and graph-level prediction accuracies than several existing models, with minor additional runtime overhead.

**Summary Of The Review:**

I think the model is reasonable both theoretically and empirically (though some recent baselines are not included for comparison). Overall, the merits outweigh the flaws, and I believe the paper is a good addition to the existing literature.

---

> ### Author Response · Authors · 2021-11-18
> **Response to Reviewer jqoo**
>
> Thank you for the insightful suggestions! Also, thanks for pointing out all these related papers on modeling joint label dependency. We have discussed these papers in the revised draft (see the blue text in related work).
>
> ----------
> **Q1: It is unclear how to extend the model to fit continuous node labels.**
>
> **Response:** It is a very good point to extend the model to fit continuous node labels. As our model is designed for discrete random variables, one straightforward solution is to discretize the continuous labels into discrete values. In the future, we will explore using more advanced methods to deal with continuous node labels.
>
> ----------
> **Q2: Insufficient baselines for comparison.**
>
> **Response:** Comparing against more methods for modeling label dependency is another good point. Among the suggested papers, CopulaGNN (Ma et al., 2021) is mainly designed for continuous and count regression tasks, and it is hard to apply the model to our setting. For LCM (Wang et al., 2021), the codes are not openly available. Therefore, we only ran experiments to compare against G3NN. More specifically, we use the same GAT as backbone for both G3NN and SMN, and we run both methods with 5 seeds. The results are presented as follows:
>
> |          |     Cora*    |   Citeseer*  |    Pubmed*   |     DBLP     |
> |:--------:|:------------:|:------------:|:------------:|:------------:|
> | G3NN-LSM | 58.60 | 51.12 | 52.44 | 79.64 |
> | G3NN-SBM | 58.28 | 50.96 | 52.20 | 78.06 |
> |    SMN   | 59.66 | 50.84 | 53.78 | 84.67 |
>
> We can see that SMN outperforms G3NN in all cases except Citeseer, which has the fewest number of average nodes (4.0) and edges (4.3) per graph (see Table 8 in appendix), making it easier to be modeled by the graph generative model of G3NN.
>
> ----------

---

### Official Review · Reviewer_RSMV · 2021-11-04

**Correctness:** 4
**Technical Novelty And Significance:** 4
**Empirical Novelty And Significance:** 3
**Recommendation:** 10
**Confidence:** 4

**Main Review:**

The technical approach exploits, in an innovative way, an insight from graphical models on the relation between pseudomarginals and the joint distribution stated in the main reference (Wainwright and Jordan 2009), to devise a formulation of a a surrogate training objective (the proxy problem) which turns out to be relatively easy to solve, leading to high predictive performance. The problem under consideration is important, and current solutions have the limitations pointed out in the paper (limited expressive capacity of CRF vs only marginal node classification for GNN models).

The paper reads very well, and hardly has any errors or inconsistencies. Information is provided at the right point, is complete and accurate. The split between main paper and supplementary material is good. The narrative and exposition flow well.

The model and algorithm descriptions are excellent.

Baselines are strong, relevant, discussed well and evaluated fairly. The experimental methodology is appropriate, well implemented, described well. A large number of analytic experiments complete the main findings. The two evaluation metrics (node and graph-level accuracy) are adequate and it is an advantage that the proposed method can optimize for either at inference time. Experimental reporting is very good, with error bars. Experimental setups are presented accurately and consistently, at an adequate level of detail. The datasets are standard and easy to access, which contributes to reproducibility.
(NB fig3 is missing a complete caption; colour coding is not explained; as a result sec5.5.7 doesn’t prove its point)

**Summary Of The Paper:**

The paper targets the task of graph node classification in the inductive setting, taking an input node and edge feature representations, and inferring a categorical label for each node. It focuses on the case where it is not sufficient to classify a node independently of its neighbours, but where information stemming from predictions for neighbouring nodes needs to be taken into account.
It improves over the current approaches by offering a performant and efficient method to combine advantages of CRFs (joint inference) with the representational power afforded by GNNs, while overcoming the computational challenges. In particular, this paper improves on the closest attempts at combining CRF and GNN, Ma+ 2018 and Qu+ 2019, which use pseudolikelihood; Qu+ 2019 is used as baseline in experiments (“GMNN”), while pseudoloikelihood as training objective is investigated in sec5.5.5.

--- update after rebuttals
I have read other reviews and the authors' replies, as well as extra experiments. I am maintaining my score.

**Summary Of The Review:**

If the scoring existed, I would give this paper a 9/10. Having to choose between 8 and 10, I have decided to go for 10, as I consider the paper to be much better than just "good".
The paper is very good under all aspects. Its contribution is clear, and the problem under consideration is real. One might argue that the innovation is slim because it consists of a single technical contribution; for this reason this might not be a game-changing paper, but it certainly improves on the state of the art by solving a tricky problem. The paper also has the merit of establishing a bridge between neural and graphical model techniques, thus encouraging further work which I expect to be fruitful.

---

> ### Author Response · Authors · 2021-11-18
> **Response to Reviewer RSMV**
>
> Thanks for your helpful comments!
>
> **Q: Further explain Fig3.**
>
> **Response:** In Fig. 3, we show three cases to illustrate that SMN can predict node labels more precisely than GNN. Specifically, each row corresponds to a case, where different colors represent different node labels. For each case, we show the prediction of GAT, edge GNN, and SMN-GAT. In all three cases, SMN-GAT correctly predicts all the node labels, so the ground-truth node labels can also be found in the last column.
>
> If we check the prediction of GAT (i.e., first column), we see that GAT makes inconsistent predictions in that some linked nodes are predicted to have different labels. For the edge GNN, it also makes incorrect predictions in the last case. By combining GAT and edge GNN together, SMN-GAT makes correct predictions in all the three cases.

---

> > ### Comment · Reviewer_RSMV · 2021-11-29
> > **Reply**
> >
> > Thank you.
> > Regarding fig.3, the caption is still unclear, it is ungrammatical, and should mention that the colour coding represents different possible labels.
> >
> > Regarding model naming: I hadn't paid much attention to the name of the model on my first read, but reviewer LsFg is right: it's a terrible name. Sticking to it, I think the paper is sure to confuse readers, reduce its reach, and in the years to come, each mention might need to be introduced by "the name given to the model is another misnomer, it should really be called (...)".

---

> > > ### Author Response · Authors · 2021-11-30
> > > **Response**
> > >
> > > Thank you for the helpful suggestions on the caption of fig.3 and the name of the model!
> > >
> > > We agree that the caption and the model name should be further revised. For now, we are not able to update the draft as the function is currently closed, but we will keep editing the paper to further improve its quality.

---

### Official Review · Reviewer_LsFg · 2021-11-04

**Correctness:** 3
**Technical Novelty And Significance:** 3
**Empirical Novelty And Significance:** 3
**Recommendation:** 8
**Confidence:** 4

**Main Review:**

Update following revision / author comments:

Thanks for clarifying and providing details on how the method compares to piecewise training. Having all of those details in the final paper would make it much stronger, and so I have raised my score.

For what it's worth, I'd suggest trying to describe the method in as general terms as possible so that people working on other structured prediction problems are more likely to learn about it and use it. The method is general and appears very promising, but currently the paper is very much written for graph problems, so I fear that researchers outside of the graph community may overlook it. Choosing a more specific name for the method could also help. The name "Structured Markov Network" is a bit too broad, as it just sounds like a Markov Network, which is a synonym for MRF. Maybe working "proxy" into the name you choose would be helpful.

Original review follows:

Strengths:

The methods are well-chosen for the task. The method is simple enough that other researchers may use it.

The empirical results are solid and interesting.

The paper is well-written.

Weaknesses:

The original piecewise training paper (Sutton & McCallum, 2009) is cited, but is not discussed in any detail. No connection is made between the proxy problem and piecewise training. I think a connection should be made and discussed, especially because the experiments do not involve actually solving the proxy problem but rather omit the marginal consistency constraint ("The last consistency constraint...can be ignored during optimization"... "We also tried some constrained
optimization methods to handle the consistency constraint, but they yield no improvement"). When refinement is not used (it is omitted in the main experiments since it doesn't consistently help) and when KL divergence is used for the divergence measure (as it is in the experiments), the actual training objective becomes even more similar to piecewise training. Piecewise training is fairly well-known in other communities like the vision community, e.g., Lin et al. (2016), so I think it's really important for this paper to draw a connection to piecewise training. I would suggest including piecewise training as a baseline to compare to, but I think that the SMN (without the marginal consistency constraint, without refinement, and when using KL) actually corresponds to a natural way to apply piecewise training to this problem.

After reading the paper, I was confused by one part of the results: Why would SMN outperform CRF? That may suggest that the approximations made during learning are beneficial, which deserves follow-up investigation. A related question is: why is refinement not helpful? The CRF-G* settings correspond to using the same models as the corresponding SMN-G* settings, but the former seek to directly solve the maximin game using loopy BP for inference during learning (I believe), rather than use the proxy problem. The SMN results are consistently better than the corresponding CRF ones. Why is the proxy problem superior? Perhaps the CRF objective is not as good for learning as the proxy problem? The proxy problem can be viewed as using "local supervision" on specific nodes and edges, which may be more learnable from supervised datasets than the traditional CRF objective which is log loss on labelings of entire graphs. Or maybe it's due to training stability: An SMN that solely solves the maximin game (is this the same as the CRF-G* models?) is described as being unstable in Sec 5.5, #5. Can you provide more details about that? Is the instability due to the use of loopy BP as the inference algorithm? How about if you pretrain the GNNs for the potentials with maximum likelihood?



Some more specific suggestions are below:

Spell out SMN ("Structured Markov Network") the first time it appears in Sec 1. I would actually also suggest changing the terminology to something more specific. "Structured Markov Network" is a pretty generic term that would evoke several kinds of existing graphical models in people's minds. E.g., some people use the term "Markov Network" to refer to undirected graphical models in general, and so the addition of the term "structured" does not really add anything since graphical models are already "structured".

The definition of GNNs in 3.2 seems unnecessarily limited. A graph neural network does not have to produce distributions over anything -- it could simply represent a graph via autoencoder training with an L2 loss, for example. Please see surveys on GNNs, such as Zhou et al. (2021), which provide a richer characterization of GNNs. If you wish to define GNN in a more constrained way for purposes of this paper, then please add "In the context of this paper, we define a GNN to be".

Sec. 3.2: "However, GNNs approximate only the marginal label distributions of nodes on training graphs, which may generalize badly and result in poor approximation of node marginal label distributions on test graphs." -- Why might they generalize badly? Would an estimate of the full label distribution be expected to generalize better? Is the paper implying here that noisy estimates of the marginals would generalize better than a noisy estimate of the joint?

Sec. 3.3: "Conditional random fields (CRFs) build graphical models for node classification." CRFs are much broader than node classification. See my comment about GNNs above.

Sec. 3.3: "intractable partition function" -- The intractability depends on the graph, right? E.g., consider chains, trees, acyclic graphs..

Sec. 4.1: "compute the a representation"

More details are needed about the DBLP dataset. What exactly is the structure of the "citation graphs" mentioned? Also, it appears that there is only a single graph in train, validation, and test -- is that correct? The test set contains papers from "after 2010" -- does that include 2010 or is it only from 2011 onward? Are the train/val/test graphs disjoint? The papers in the test set will mostly be citing papers from the training and validation sets, or are the latter removed from the test graph to avoid overlapping nodes among splits?

References:

Guosheng Lin, Chunhua Shen, Anton van dan Hengel, Ian Reid. Efficient piecewise training of deep structured models for semantic segmentation. CVPR 2016.

Jie Zhou, Ganqu Cui, Shengding Hu, Zhengyan Zhang, Cheng Yang, Zhiyuan Liu, Lifeng Wang, Changcheng Li, Maosong Sun. Graph Neural Networks: A Review of Methods and Applications. 2021.



**Summary Of The Paper:**

This paper proposes a CRF for classifying nodes in graphs where the CRF has a potential for each node and edge in the graph. GNNs are used for computing the potentials, one GNN for node potentials and one GNN for edge potentials. For general graphs, computing the partition function is intractable, so approximations are used during both learning and inference.  Learning draws from prior work and combines learning pseudomarginals of nodes and edges with GNNs and optionally some steps of "refinement" by optimizing a maximin game equivalent to likelihood maximization. Inference uses sum-product or max-product loopy BP (they perform similarly, though max-product is slightly better). The procedure is called a "Structured Markov Network" (SMN). Experimental results on several graph node classification tasks show that SMNs outperform several baselines, including both CRFs with standard training (using approximate inference during training) and CRFs trained with pseudolikelihood.


**Summary Of The Review:**

While the paper is well-written and the results show consistent improvements, the paper is lacking in terms of any connection made to piecewise training (which the best version of the SMN essentially boils down to, as far as I can tell) and an analysis of why optimizing the proxy problem is superior to the original learning problem. These are potentially fixable issues.

---

> ### Author Response · Authors · 2021-11-18
> **Response to Reviewer LsFg (1)**
>
> Thanks for the insightful comments!
>
> ----------
>
> ### Q1: The connection between our method and piecewise training.
>
> **Response:** It is a great point to draw connections to piecewise training, as piecewise training is a useful tool for training CRFs and is well-recognized in the CV domain.
>
> **(1) Major difference between our method and piecewise training.**
>
> Both our method and piecewise training can be understood as trying to train a local model on each node $s$ and edge $(s,t)$ to predict their labels, i.e., $\tau_s(y_s)$ and $\tau_{st}(y_s, y_t)$. In this sense, both methods share similar ideas.
>
> The major difference lies in how we combine those local pieces on nodes and edges into a joint distribution. Formally, as in Eq. (2) of our draft, the joint distribution of pair-wise CRFs can be formulated as $p(\mathbf{y}_V | \mathbf{x}_V) \propto \exp (\sum \theta(y_s) + \sum
>  \theta(y_s, y_t)) $, where we need to specify the functions $\theta(y_s) $ and $\theta(y_s, y_t) $.
>
> - Piecewise Training: In piecewise training, the functions are essentially specified as $\theta_s(y_s) = \log \tau_s(y_s)$ and $\theta_{st}(y_s, y_t) = \log \tau_{st}(y_s, y_t)$.
>
> - Our method: As in Eq. (5), our method sets the parameters as $\theta_s(y_s) = \log \tau_s(y_s)$ and $\theta_{st}(y_s, y_t) = \log \tau_{st}(y_s, y_t) - \log \tau_s(y_s) - \log \tau_t(y_t)$.
>
> **(2) Theoretical advantage of our method over piecewise training.**
>
> We first discuss the theoretical advantage of our method. As explained in Sutton et al. (2009), piecewise training essentially maximizes a lower bound of the likelihood function on training data, but "If the graph is connected, however, then the bound is not tight, and in practice it is extremely loose." This implies that the joint distribution learned by piecewise training often cannot well fit training data.
>
> In contrast, by solving the proxy problem as in our method, the moment-matching conditions are roughly satisfied, meaning that the joint distribution learned by our method can better fit training data.
>
> For example, on tree-structured graphs, if $\tau_s(y_s)$ and $\tau_{st}(y_s, y_t)$ can perfectly fit the local node/edge marginals on training data, then the joint distribution learned by our models can perfectly fit the training data, whereas the joint distribution formed by piecewise training cannot. To better illustrate that, let's consider a toy case where a graph only has two linked nodes, and each node has two possible labels 0 and 1. Suppose the ground-truth joint label distribution $p(y_s, y_t)$ is as follows:
>
> |             |  y_t = 0 |  y_t = 1 |
> |:-----------:|:--------:|:--------:|
> | **y_s = 0** |   0.34   |   0.20   |
> | **y_s = 1** |   0.36   |   0.10   |
>
> Suppose also that $\tau_s(y_s)$, $\tau_s(y_s)$, and $\tau_{st}(y_s, y_t)$ can perfectly fit the marginal distributions on nodes and edges, i.e., $\tau_s(y_s) = p(y_s)$, $\tau_t(y_t) = p(y_t)$, and $\tau_{st}(y_s, y_t) = p(y_s, y_t)$.
>
> Then the joint distribution learned by our method with these local components is:
>
> |             |  y_t = 0 |  y_t = 1 |
> |:-----------:|:--------:|:--------:|
> | **y_s = 0** |   0.34   |   0.20   |
> | **y_s = 1** |   0.36   |   0.10   |
>
> This is the same as the ground-truth joint label distribution, and the label configuration $(y_s = 1, y_t = 0)$ gets the highest probability, which agrees with the ground-truth data distribution.
>
> For piecewise training, the joint distribution is computed as:
>
> |             |  y_t = 0 |  y_t = 1 |
> |:-----------:|:--------:|:--------:|
> | **y_s = 0** |   0.44   |   0.11   |
> | **y_s = 1** |   0.40   |   0.05   |
>
> We can see that this distribution is quite different from the ground-truth joint distribution, and the most likely label configuration is $(y_s = 0, y_t = 0)$, which is inconsistent with the ground-truth data. Therefore, we see that piecewise training does not fit training data well, which is undesirable.
>
> **(3) Empirical comparison of our method and piecewise training.**
>
> We also compare against piecewise training empirically on four datasets with the same GNN backbone (GAT). The results over 10 different runs are presented as follows:
>
> |                    |     Cora*    |   Citeseer*  |    Pubmed*   |     DBLP     |
> |:------------------:|:------------:|:------------:|:------------:|:------------:|
> | Piecewise Training | 57.08 | 48.04 | 51.95 | 82.93 |
> |    SMN   | 58.78 | 49.02 | 52.91 | 84.84 |
>
> We can see that our method achieves consistently better results.

---

> > ### Comment · Reviewer_LsFg · 2021-11-27
> > **reply**
> >
> > Interesting, thanks!

---

> ### Author Response · Authors · 2021-11-19
> **Response to Reviewer LsFg (2)**
>
> ----------
>
> ### Q2: Comparison with CRF.
>
> **Response:** Further comparing our method with CRF is another insightful suggestion!
>
> **(1) Why would SMN outperform CRF? Why is the proxy problem superior?**
>
> Both classical CRF and our method try to maximize data likelihood for model training, and the difference lies in how we achieve that.
>
> - CRF: The typical way for likelihood maximization is to formalize the problem as a maximin game or a saddlepoint problem, as elaborated in Eq. (3) of Sec. 3.3. Then the problem can be optimized by alternating between loopy BP and gradient descent. However, such a maximin game is often hard to optimize by nature due to training instability, which is also observed in training generative adversarial networks (GANs). This is illustrated in Fig. 2 of our draft, where we can see that the training curve of CRF is very unstable. This training instability is not caused by loopy BP, but by the nature of the maximin game.
>
> - Our Method: Our method performs likelihood maximization in a different way. We investigate the optimality condition that a model with maximum likelihood should satisfy. Then we design a proxy problem whose solution roughly satisfies the optimality condition, and thus this proxy problem can be viewed as a surrogate for likelihood maximization. The major attractive property is that optimizing this proxy problem only requires "local supervision" on nodes and edges, and thus our method is much more stable in terms of training, achieving much better results.
>
> Therefore, although CRF and our method both try to maximize data likelihood, our method has better training stability, yielding better results.
>
> Besides, another difference is that we parameterize the joint distribution in a special way, i.e., Eq. (5) of Sec. 4, which allows our approach to have better theoretical properties as mentioned in Prop. 1.
>
> **(2) Why is refinement not helpful?**
>
> The refinement process further optimizes model parameters using the maximin game. Nevertheless, as aforementioned, solving the maximin game is unstable. Besides, the joint distribution derived by solving the proxy problem is already very close to the optimal solution. Because of these two reasons, refinement might not be very helpful in some cases. In our experiments, only on datasets with a large number of training data (e.g., PPI-10, PPI) does refinement further improve the final results (see Tab. 4).
>
> ----------
>
> ### Q3: There are some writing issues, i.e., SMN, definition of GNNs, and CRFs
> **Response:** Thank you for the helpful suggestions! We have fixed the writing issues in the revised draft, including spelling out SMN the first time it appears, revising the definitions of GNNs and CRFs in Sec. 3.2 and Sec. 3.3, and fixing the typos (see the light blue text).
>
> ----------
>
> ### Q4: The generalization abilities of GNNs
> **Response:** We are sorry about the confusion. GNNs only try to estimate the marginal distributions on nodes. In contrast, modeling the joint distribution requires two models to estimate the marginal distributions on nodes and edges respectively. Intuitively, these two models for node marginals and edge marginals can mutually correct each other, resulting in better generalizability. Nevertheless, we are unaware of any theoretical proof of that.
>
> ----------
>
> ### Q5: About the DBLP Dataset
> **Response:** We are sorry that some details of the DBLP dataset were missing due to space limits. Below is the detailed description of the dataset.
>
> We build the DBLP dataset from the DBLP citation network. Scientific papers from eight conferences are treated as nodes, and we further split them into three categories for classification according to conference domains. For each paper, we compute the mean GloVe embedding of words in the title and abstract as node features.
>
> We split the dataset into three disjoint graphs for training/validation/test.The training graph contains papers published before 1999 (with 1999 included). The validation graph contains papers published between 2000 and 2009 (with 2000 and 2009 included). The test graph contains papers published after 2010 (with 2010 included). There exists an undirected edge between two papers if one paper cites the other one. Cross-graph edges (e.g., an edge between a paper in the training set and a paper in the validation set) are removed.
>
> We have updated the draft accordingly (see the light blue text in the Sec. F.2).
>
> ----------

---

> > ### Comment · Reviewer_LsFg · 2021-11-27
> > **response**
> >
> > Thanks for addressing these and clarifying!
> >
> > I did still have a question though. You mention: "This training instability is not caused by loopy BP, but by the nature of the maximin game." To make this claim, I think you would need to use a theoretical argument or run experiments showing that the instability remains no matter what inference algorithm is used. The draft notes that MCMC can be used (Sec. 3.3 states: "This can be done by MCMC, but the time cost is high, so approximate inference is often used, such as loopy belief propagation (Murphy et al., 1999).") but only loopy BP is used in the experiments. How then can one be sure that the training instability is not caused by loopy BP if you only tried to solve the maximin game when using loopy BP? Might it be more stable with MCMC?

---

> > > ### Author Response · Authors · 2021-11-28
> > > **Response**
> > >
> > > Thank you for the follow-up questions and suggestions!
> > >
> > > Using MCMC (e.g., Gibbs sampling) to replace loopy BP for optimization is an interesting suggestion. Nevertheless, MCMC requires a burn-in period, which can be expensive. To solve the problem, one idea is to leverage k-step contrastive divergence, which is an MCMC method used for optimizing restricted Boltzmann machines. Basically, the method runs Gibbs sampling for only a few steps to speed up optimization. We did additional comparison against this method on Cora*, Citeseer*, and Pubmed*, where GAT was used as the backbone model and k was set to 10. The results are presented as follows:
> > >
> > > |            | Cora* | Citeseer* | Pubmed* |
> > > |:----------:|:-----:|:---------:|:-------:|
> > > | 10-step CD |  53.08 |    44.18   |   49.04  |
> > > |     SMN    |  58.78 |    49.02   |   52.91  |
> > >
> > > We can see that MCMC still gets worse results. The reason might be that we only ran MCMC for 10 steps for efficiency purposes, and thus the samples were not good enough, yielding inferior results. Despite that, we do agree that combining with MCMC would be a promising direction to further improve optimizing CRFs, and we will further explore how to address the efficiency issue of MCMC in our context.

---

### Official Review · Reviewer_DQ8r · 2021-11-05

**Correctness:** 4
**Technical Novelty And Significance:** 2
**Empirical Novelty And Significance:** 3
**Recommendation:** 8
**Confidence:** 3

**Main Review:**

Strengths
* Paper is well written
* The proposed method is a novel contribution that combines ideas from CRFs and GNNs.

Weaknesses
* I consider the experiment section to be a minor weakness. I understand that the method sits in between CRFs and GNNs, but it would be great to compare against methods beyond those from CRFs and GNNs, e.g., SSVMs and others.
* Another minor weakness. The technical contribution is limited as it builds on old ideas from graphical models. My understanding is that the key to efficiency relies on solving the proxy problem, which, as the authors stated, is an idea that dates back to at least the early 2000s in the context of graphical models.


**Summary Of The Paper:**

Authors study the problem of node labeling in the inductive case, i.e., at test time the goal is to label all the nodes of a given graph.
For that problem, several variants of GNNs and CRFs have been proposed in the past, and the authors propose a Structural Markov Network that uses GNNs to model the potential functions of a CRF with the subtle difference that a proxy optimization problem is solved to make learning more efficient. Experiments are provided to demonstrate the applicability of their method.


**Summary Of The Review:**

In my opinion, the paper contains good contributions that are worth publishing at ICLR. While I think the experiments section could benefit from additional comparisons, the current state of the empirical evaluation is reasonable. A score of 7 would reflect better my evaluation of this work as I find the technical contributions to be okay but somewhat limited.

---

> ### Author Response · Authors · 2021-11-18
> **Response to Reviewer DQ8r**
>
> **Q: Additional comparison to other methods, e.g., SSVMs.**
>
> **Response:** Thanks for the helpful suggestions! As you pointed out, we only compared against GNN and CRF. Following your suggestions, we added additional comparison against SSVM, and the results are presented as follows:
>
> |      | Cora* | Citeseer* | Pubmed* | DBLP |
> |:----:|:-----:|:---------:|:-------:|:----:|
> | SSVM |  42.7 |    33.2   |   43.2  | 51.5 |
> |  SMN |  58.8 |    49.0   |   52.9  | 84.8 |
>
> We can see that our approach SMN achieves better results than SSVM, as SMN has higher model capacities.

---

### Public Comment · ~Zijing_Ou1 · 2022-03-19
**difference between CRF and SPN w/o proxy**

Dear authors,

Thanks for your excellent work. It is exactly interesting to combine the idea of CRF and GNN.

I have some questions about figure 2. What is the difference between CRF and SPN w/o proxy? It seems that both two methods train the model via the maximin game in Eq.3 with loopy belief propagation.

Besides, would you plan to release the code?

---

> ### Public Comment · ~Meng_Qu2 · 2022-03-20
> **Response**
>
> Thanks for your interest in our paper!
>
> CRF and SPN w/o proxy differ in how they parameterize the theta-functions. For CRF, we directly parameterize the theta-functions as the logits computed by GNNs. For SPN w/o proxy, we parameterize the theta-functions according to Eq. (5) of our paper, where GNNs are used to compute the tau-functions (pseudomarginals), which are further combined into the theta-functions.
>
> We are now finalizing the code. Once done, we will upload the code to https://github.com/DeepGraphLearning/SPN. Stay tuned!

---

> > ### Public Comment · ~Zijing_Ou1 · 2022-03-31
> > **Question on node-level inference**
> >
> > Thanks for your reply! It has addressed my questions above. But here I have another follow-up question.
> >
> > In the inference phase, say the node-level one specifically here, you propose to run loopy BP for the estimate of the marginal $p_\theta (y_s )$. Why not directly apply the output of proxy networks? I mean according to Proposition 1, the pseudomarginal is approximately equal to the marginal of loopy BP. In practice, it seems that using the pseudomarginals in inference phase is more attractive than running loopy BP since the latter is time-consuming.

---

> > > ### Public Comment · ~Huiyu_Cai1 · 2022-04-01
> > > **Response**
> > >
> > > Hi Zijing. Thanks for your question!
> > >
> > > Indeed, we could directly infer the node labels based on the output of the node proxy network. But by doing that we would have thrown away the CRF, as the node proxy network $p_\theta(y_s)$ is trained by standard cross entropy loss.
> > >
> > > The power of CRF lies in the joint label distribution $p_\theta(\mathbf{y}_V)$. The goal of max-product loopy BP is to infer the best node labels from this *joint* distribution. This set of best labels may differ from what you'd get by taking the best label from each node *marginal* distribution, which can not model the label dependencies between nodes.
> > >
> > > For sum-product loopy BP, the goal is to infer the marginal label distribution on each node. During training, we implicitly force $\sum_{y_s} \tau_{st}(y_s, y_t) = \tau_t(y_t)$ (through the proxy objectives), and thus the marginal distribution inferred by sum-product loopy BP would be very similar to the node-wise pseudomarginal on training graphs. But during inference, the constraint may not hold on test graphs, which means that those $\tau$ functions no longer satisfy the conditions in proposition 1, and thus the marginal distribution inferred by sum-product loopy BP can also be different from the pseudomarginal on each node.

---

> > > > ### Public Comment · ~Zijing_Ou1 · 2022-04-01
> > > > **Response**
> > > >
> > > > Hi, thanks for your super quick reply!
> > > >
> > > > It is true that for sum-product loopy BP, the $\tau$ functions no longer satisfy the conditions in proposition 1, and the marginal inferred by loopy BP would be somewhat different from the pseudomarginal.
> > > > But I am curious about the practice perspective. I mean during training, the cross-entropy loss is directly applied on top of $\tau$ functions. To some extent, $\tau$ looks like the predictive logits of the target category distribution (not exactly true, but practically we could view it from this perspective). Therein, I suspect the experimental results would be good if directly using $\tau$ to predict the label for each node (maybe you should scale $\tau$ to a valid density during testing). Did you conduct relevant experiments to verify this hypothesis?

---

> > > > > ### Public Comment · ~Meng_Qu2 · 2022-04-04
> > > > > **Response**
> > > > >
> > > > > Thanks for following up!
> > > > >
> > > > > If I understood correctly, you meant we can direct treat the $\tau$ function of each node as logits for label prediction? If this is the case, then it is exactly the GNN baseline in our experiment. You might check Table 1 and Table 2 for the detailed comparison.
> > > > >
> > > > > By the way, the code of SPN is now available at https://github.com/DeepGraphLearning/SPN.

---

### Decision · Program_Chairs · 2022-01-20

**Decision:**

Accept (Oral)

**Comment:**

Most of the existing GNN based methods model the node labels independently and ignore the joint dependency of node labels. The CRF-based methods work in this setting, but they are hard to learn. Hence, this paper proposes to ease the learning difficulty by solving the proxy problem and simplifying the max-min problem.

The SMN model proposed in this work is much cheaper than the CRF method. For parameters, since the node GNN and edge GNN share parameters in layers, only a few amounts extra parameters are introduced. As for the training time, it just doubles the general GNNs. Compared with CRF methods, the cost saved by SMN is significant.

Empirically, SMN works well in most settings, in terms of both node-level accuracy and graph-level accuracy, the different backbones, and different datasets. Meanwhile, the authors provide results to show the effect of refinement, the shared GNNs, the different learning methods, convergence, and a tiny case study. The experimental results are significant and well organized.

After the rebuttal and discussion, all reviewers are in a favor of accepting this submission.